# CONTEXTUAL CAUSAL BAYESIAN OPTIMISATION

**Vahan Arsenyan**
CREST, ENSAE, IP Paris, FR

**Antoine Grosnit**
Huawei Noah's Ark Lab, UK

**Haitham Bou-Ammar**
Huawei Noah's Ark Lab, UK

**Arnak Dalalyan**
CREST, ENSAE, IP Paris, FR

## ABSTRACT

We introduce a unified framework for contextual and causal Bayesian optimisation, which aims to design intervention policies maximising the expectation of a target variable. Our approach leverages both observed contextual information and known causal graph structures to guide the search. Within this framework, we propose a novel algorithm that jointly optimises over policies and the sets of variables on which these policies are defined. This thereby extends and unifies two previously distinct approaches: Causal Bayesian Optimisation and Contextual Bayesian Optimisation, while also addressing their limitations in scenarios that yield suboptimal results. We derive worst-case and instance-dependent high-probability regret bounds for our algorithm. We report experimental results across diverse environments, corroborating that our approach achieves sublinear regret and reduces sample complexity in high-dimensional settings.

## 1 INTRODUCTION

Bayesian optimisation (Garnett, 2023) is a powerful approach for optimising expensive black-box functions, often formulated as conditional expectations of a target variable given that intervenable variables are assigned some specific values. In settings where intervenable variables exhibit causal relationships, specialised Causal Bayesian Optimisation (CaBO) algorithms can achieve superior performance by strategically focusing on select subsets of variables (Lee & Bareinboim, 2018). The critical challenge is to identify these subsets efficiently, since an exhaustive search is computationally intractable and sample inefficient.

On the one hand, existing CaBO research develops specific tools to reduce regret by leveraging the causal graph assumed to be known. Consistent with prior work (Aglietti et al., 2020b; Zhang & Bareinboim, 2022), we restrict attention to precise interventions (Pearl, 2009, Chapter 3) and assume the graph is given. Although extensive work on causal discovery provides methods for inferring such graphs (Spirtes et al., 2000; Hoyer et al., 2008; Hyttinen et al., 2013; Duong et al., 2025), this direction lies beyond the scope of our study. On the other hand, Contextual Bayesian Optimisation (CoBO) leverages observable context variables but is constrained by a fixed policy scope, that is, a fixed tuple of intervenable and contextual variables. As we demonstrate in the example in Section 2, this fixed scope constraint can lead to provably linear regret, which is suboptimal.

Optimising over multiple policy scopes arises in several domains. In complex environments such as video games, fully specifying the state and action spaces is intractable, both computationally and from an optimisation standpoint, so practitioners adopt alternative design choices (Berner et al. (2019), Vinyals et al. (2019)), each inducing a distinct policy scope. In portfolio optimisation, multiple strategies are proposed, each controlling particular sets of portfolio parameters conditioned on market observables (Cakmak & Özekici (2006), Cakmak et al. (2020)). In neural architecture search (Elsken et al. (2019)), multiple strategies exist: one may monitor gradient and activation magnitudes and adjust network width and weight initialisation accordingly; another may track test-time performance and the generalisation gap, adjusting learning rate, gradient clipping, and depth. While jointly controlling all architectural parameters and conditioning on all performance metrics might seem optimal, the resulting search space is infeasible, so researchers adopt heuristic, scope-specific strategies. These examples fit naturally within our framework of optimising over multiple scopes.

The need for our proposed framework emerges from three critical observations discussed above: (1) overcoming linear regret in CoBO requires extending beyond fixed policy scopes, (2) the exponential growth of possible policy scopes with respect to the number of variables makes exhaustive search computationally intractable, and (3) while causal relationships can inform policy scope selection, existing CaBO approaches do not incorporate contextual variables. In response, we introduce a unified framework for contextual and causal Bayesian Optimisation (BO).

Our framework, termed CoCa-BO for Contextual-Causal Bayesian Optimisation, is designed to reduce the search space of policy scopes and guarantee convergence to the optimal policy. It addresses the challenges of applying the causal acquisition function (Aglietti et al., 2020b) for policy selection in contextual settings. We present analytical cases where existing methods fail, and show both theoretically and experimentally that CoCa-BO converges to the optimum. Experimental results further demonstrate that CoCa-BO achieves lower sample complexity in large-scale systems and outperforms existing methods.

Below, we summarise our main contributions:
- We propose a novel method for context-aware causal Bayesian optimisation that addresses policy selection in context-aware settings with multiple possible policy scopes.
- We provide a general, scalable, and practical framework for implementing causality-aware optimisation algorithms, along with a suite of environments for evaluation and benchmarking.
- Assuming Gaussian process priors, we establish both worst-case and instance-dependent, high-probability upper bounds on the regret of our method. These bounds are valid under mild assumptions on the kernels of the Gaussian process priors and highlight the impact of the number of steps on the cumulative regret.

**Notation** Random variables are denoted by capital letters like $X$ and their values by corresponding lowercase letters like $x$. The domain of random variable $X$ is denoted $\mathfrak{X}_X$. Sets of variables are denoted by bold capital letters and the set of corresponding values by bold lowercase letters. Vectors are always considered as 1-column matrices and for any vector $\mathbf{x}$, we denote by $\mathbf{x}^\mathsf{T}$ its transpose. Identity matrices are denoted by $\mathbf{I}$ without indicating the dimension, which is clear from the context. We denote by $\mathbb{N}^*$ the set of strictly positive integers, and by $[m]$ the set of strictly positive integers smaller than, or equal to, $m \in \mathbb{N}^*$. For $\mathbf{x} \in \mathbb{R}^m$, $\|\mathbf{x}\|_1 = \sum_{i=1}^m |x_i|$ and $\|\mathbf{x}\|_\infty = \max_{1 \le i \le m} |x_i|$.

## 2 BACKGROUND

This section aims to provide a background on possibly optimal mixed policy scopes and on their importance for the contextual decision making setup where the causal graph is available.

**Preliminaries** We consider the following general problem: in the absence of any intervention, the environment generates a random vector $\mathbf{V}$ according to a probability distribution denoted by $P_{\mathbf{V}}^0$ on $\mathfrak{X}_{\mathbf{V}}$. We are allowed to perform certain interventions (a precise definition will be given later) on the components of $\mathbf{V}$, which result in modifications of their joint distribution. Each set of admissible interventions $\pi$, referred to as a *policy*, induces a new distribution $P_{\mathbf{V}}^\pi$ on $\mathfrak{X}_{\mathbf{V}}$. $\Pi$ denotes the set of all admissible policies; it will be specified below and may be infinite and even uncountable in general.

We assume that our objective is to identify a policy that leads to a nearly maximal expected reward. More precisely, for a given reward function $r : \mathfrak{X}_{\mathbf{V}} \to \mathbb{R}$, our goal is to find a policy $\hat{\pi}$ such that

$$\mu(\hat{\pi}) := \mathbb{E}_{P_{\mathbf{V}}^{\hat{\pi}}}[r(\mathbf{V})],$$

the expected reward under $\hat{\pi}$, is close to the optimal expected reward $\sup_{\pi \in \Pi} \mathbb{E}_{P_{\mathbf{V}}^\pi}[r(\mathbf{V})]$. We will focus on the case where $r(\mathbf{V}) = \langle e, \mathbf{V} \rangle$, where $e$ is an element of the canonical basis. This is equivalent to assuming that $Y := r(\mathbf{V})$ is a component of $\mathbf{V}$.

To define what a policy is and how the set of admissible policies is specified, we follow (Lee & Bareinboim, 2020). Assume that $P_{\mathbf{V}}^0 = P_{\mathbf{V}}^{\mathfrak{M}}$ is given by a structural causal model $\mathfrak{M}$ (Pearl, 2009, Chapter 7). A structural causal model (SCM) $\mathfrak{M}$ is a quadruple $(\mathbf{V}, \mathbf{U}, \mathcal{F}, P_{\mathbf{U}})$, where $\mathbf{V}$ and $\mathbf{U}$ are, respectively, a set of observed and latent variables. $\mathcal{F}$ is a collection of functions such that for each $V \in \mathbf{V}$, there exists $f_V \in \mathcal{F}$ that determines the value of $V$ based on other variables in $\mathbf{V} \cup \mathbf{U}$. That is, for some $\mathbf{PA}_V \subseteq \mathbf{V} \setminus \{V\}$ and $\mathbf{U}_V \subseteq \mathbf{U}$, $V = f_V(\mathbf{PA}_V, \mathbf{U}_V)$. The latent variables $\mathbf{U}$ are jointly distributed according to $P_{\mathbf{U}}$. Consequently, the SCM $\mathfrak{M}$ induces a joint distribution $P_{\mathbf{V}}^{\mathfrak{M}}$ over the observed variables $\mathbf{V}$.

The SCM $\mathfrak{M}$ also defines a directed acyclic graph (DAG) $\mathcal{G}$ with two types of nodes. First-type nodes represent variables from $\mathbf{V}$, while second-type nodes represent variables from $\mathbf{U}$. Only first-type nodes have incoming edges. An edge from a node corresponding to a variable $Z$ to a first-type node corresponding to $V$ indicates that $Z \in \mathbf{PA}_V \cup \mathbf{U}_V$. The policies we consider in this paper are defined on mixed policy scopes (MPS), which should be compatible with the graph $\mathcal{G}$ of the SCM $\mathfrak{M}$. The following definitions and terminology are based on (Lee & Bareinboim, 2020).

**Definition 1** (Mixed policy scope). Let $\mathbf{X}^* \subseteq \mathbf{V} \setminus \{Y\}$ and $\mathbf{C}^* \subseteq \mathbf{V} \setminus \{Y\}$ be two sets, termed the set of intervenable variables and the set of contextual variables, respectively. A *mixed policy scope* $\mathcal{S}$ compatible with $\mathcal{G}$ is defined as a collection of pairs $(X; \mathbf{C}_X)$ such that $X \in \mathbf{X}^*$, $\mathbf{C}_X \subseteq \mathbf{C}^* \setminus \{X\}$, and the following compatibility condition is satisfied. The graph $\mathcal{G}_\mathcal{S}$ obtained from $\mathcal{G}$ by removing edges into $X$ and adding new ones from $\mathbf{C}_X$ to $X$, for every $(X; \mathbf{C}_X) \in \mathcal{S}$, remains acyclic.

We denote the set of all MPSs compatible with $\mathcal{G}$ by $\mathbb{S}[\mathcal{G}]$. The notation $\mathbf{X}(\mathcal{S})$ designates the set of intervenable variables for an MPS $\mathcal{S}$, and $\mathbf{C}(\mathcal{S})$ the contextual ones.

**Definition 2.** A *mixed policy* $\pi$ *based on* $\mathcal{S}$ is a collection $\pi = \{\pi_{X|\mathbf{C}_X}\}_{(X;\mathbf{C}_X)\in\mathcal{S}}$, where $\pi_{X|\mathbf{C}_X}$ is a mapping from $\mathfrak{X}_{\mathbf{C}_X}$ to $\mathfrak{X}_X$.

For every MPS $\mathcal{S} \in \mathbb{S}[\mathcal{G}]$, the set of all policies based on $\mathcal{S}$ is denoted by $\Pi_\mathcal{S}$. We define $\Pi_\mathcal{G} = \bigcup_{\mathcal{S}\in\mathbb{S}[\mathcal{G}]} \Pi_\mathcal{S}$ as the space of all policies.

If $\mathcal{G}$ is the causal graph of an SCM $\mathfrak{M} = (\mathbf{V}, \mathbf{U}, \mathcal{F}, P_\mathbf{U})$, every mixed policy $\pi$ based on $\mathcal{S} \in \mathbb{S}[\mathcal{G}]$ defines a new SCM $\mathfrak{M}^\pi = (\mathbf{V}, \mathbf{U}, \mathcal{F}', P_\mathbf{U})$ with $\mathcal{F}'$ obtained as follows. For each intervenable variable $X \in \mathbf{X}(\mathcal{S})$, replace the function $f_X$ from $\mathcal{F}$ by the new function $\pi_{X|\mathbf{C}_X} : \mathfrak{X}_{\mathbf{C}_X} \to \mathfrak{X}_X$. This new SCM $\mathfrak{M}^\pi$ induces a new distribution on $\mathbf{V}$, denoted by $P_\mathbf{V}^\pi$.

---

The agent is given the graph $\mathcal{G}$ of an unknown SCM $\mathfrak{M}$. At each step $t = 1, 2, \ldots, T$

    a) the agent chooses a mixed policy scope $\mathcal{S}_t \in \mathbb{S}[\mathcal{G}]$ and a mixed policy $\pi_t \in \Pi_{\mathcal{S}_t}$, based on the past observations $\mathcal{D}^{t-1} = (\mathcal{S}_\ell, \pi_\ell, \mathbf{c}_\ell, y_\ell)_{1\leqslant\ell\leqslant t-1}$,

    b) the environment generates $\mathbf{v}_t$ according to the distribution $P_\mathbf{V}^{\pi_t}$,

    c) the agent observes the components $(\mathbf{c}_t, y_t)$ of $\mathbf{v}_t$, corresponding to contextual variables of $\mathcal{S}_t$ and the reward.

The agent aims to minimise the regret $R_T = T \sup_{\pi\in\Pi_\mathcal{G}} \mu(\pi) - \sum_{t=1}^T \mu(\pi_t)$, with $\mu(\pi) = \mathbb{E}_{P_\mathbf{V}^\pi}[Y]$.

---

**Possibly-optimal mixed policy scopes**     Our work focuses on a special set of MPSs that are possibly optimal under an SCM compatible with $\mathcal{G}$ (Lee & Bareinboim, 2020). Let $\mu_\mathcal{S}^* = \sup_{\pi\in\Pi_\mathcal{S}} \mu(\pi)$ be the largest expected reward corresponding to the MPS $\mathcal{S}$.

**Definition 3** (Possibly-Optimal MPS (POMPS)). Given $(\mathcal{G}, Y, \mathbf{X}^*, \mathbf{C}^*)$, we say that $\mathcal{S} \in \mathbb{S}[\mathcal{G}]$ is a possibly-optimal MPS if there exists an SCM $\mathfrak{M}$ compatible with the graph $\mathcal{G}$ such that $\mu_\mathcal{S}^* > \mu_{\mathcal{S}'}^*$ for any $\mathcal{S}' \in \mathbb{S}[\mathcal{G}]$ different from $\mathcal{S}$[1].

We denote by $\mathbb{S}^*[\mathcal{G}]$ the set of all POMPSs compatible with $\mathcal{G}$. The policy considered in this work selects $\mathcal{S}_t$ from $\mathbb{S}^*[\mathcal{G}]$. While further restricting the candidate set could reduce computational cost, it would risk linear regret: for any $\mathcal{S} \in \mathbb{S}^*[\mathcal{G}]$, there exists an SCM $\mathfrak{M}$ compatible with $\mathcal{G}$ for which $\mathcal{S}$ is optimal. Thus, given only $\mathcal{G}$, the agent cannot exclude any POMPS from the set of candidate scopes without potentially discarding the optimal policy and incurring linear regret.

**Example of CoBO/CaBO's failure modes**     Let us illustrate through an example the role of incorporating both causal and contextual information. Specifically, we demonstrate that neglecting either component can lead to regret that is not sublinear in the number of steps $T$. Consider the example in Figure 1, where the context variable $C$ influences both the intervenable variable $X_2$ and the outcome $Y$. The causal graph $\mathcal{G}$ also includes the intervenable variable $X_1$, which affects $X_2$.

First, we observe that there exists a policy achieving an expected value of $Y$ equal to $1/3$. Specifically, consider the mixed policy $\pi^{(1)}$ defined on the MPS $\mathcal{S}_1 = \{(X_1; C)\}$ by $\pi_{X_1|C}^{(1)}(c) = -c$. Under this policy, we have $\mathbb{E}_{\pi^{(1)}}[Y] = \mathbb{E}_{\pi^{(1)}}[U_2^2] \cdot \mathbb{E}_{\pi^{(1)}}\left[e^{-(X_1+C)^2}\right] + \mathbb{E}_{\pi^{(1)}}[C] = \mathbb{E}[U_2^2] = 1/3$.

---

[1]$\mathcal{S}$ must additionally be non-redundant under optimality. Please see Definition 4 of Lee & Bareinboim (2020).

$$U_1, U_2 \sim \text{Uniform}([-1, 1])$$
$$X_1 = U_1, \quad C = U_1$$
$$X_2 = U_2 \exp\{-(X_1 + C)^2\}$$
$$Y = U_2 X_2 + C$$

(a) Graph $\mathcal{G}$        (b) Structural causal model

Figure 1: An SCM and its DAG $\mathcal{G}$ for which CoBO and CaBO fail to converge to the optimum.

If we ignore the contextual information and rely solely on the causal relationships, at each step the choice reduces to one of three options: no intervention, intervention on $X_1$, or intervention on $X_2$. Intervening simultaneously on both $X_1$ and $X_2$ is excluded because, according to the graph $\mathcal{G}$ depicted above, intervening on $X_2$ renders the value of $X_1$ irrelevant to $Y$. In the case of no intervention, we have $\mathbb{E}[Y] = 1/3\, \mathbb{E}\big[e^{-4U_1^2}\big] \leqslant 1/6$. If we intervene on $X_2$, it becomes independent of $U_2$ and the expected value of $Y$ reduces to zero. Finally, intervening on $X_1$ by setting it to a fixed value $x$ yields $\mathbb{E}_{\text{do}(X_1=x)}[Y] = 1/3\, \mathbb{E}\big[e^{-(x+U_1)^2}\big] \leqslant 1/3\, \mathbb{E}\big[e^{-U_1^2}\big] < 1/4$. Therefore, for every policy $\pi$ that does not incorporate the context, the expected reward is strictly less than $1/4$, and consequently, the cumulative regret of any such algorithm is at least $T/12$.

Finally, suppose we ignore the causal structure and rely solely on the contextual information. Then, the decision reduces to selecting a function $\pi^{(2)} : c \mapsto (x_1, x_2)$ that specifies the values of $(X_1, X_2)$ given $C = c$. Under such a policy, $\mathbb{E}_{\pi^{(2)}}[Y] = \mathbb{E}_{\pi^{(2)}}[U_2 X_2 + C] = \mathbb{E}\big[U_2\, \pi_2^{(2)}(C) + C\big]$. Note that $\mathbb{E}[U_2] = \mathbb{E}[C] = 0$, and $\pi_2^{(2)}(C)$ being a deterministic function of $C$, is independent of $U_2$. Therefore, $\mathbb{E}_{\pi^{(2)}}[Y] = 0$. This shows that, for any context-only policies used across the $T$ steps, the expected reward is zero. Thus, neglecting the causal structure leads to a regret of at least $T/3$.

## 3   PROPOSED ALGORITHM

In this section, we provide high-level description of the identification of intervenable scopes and the selection of optimal policies. We then formalise the contextual BO strategy used within each scope. Finally, we discuss why CaBO's POMPSs selection strategy is inadequate in contextual settings and motivate our alternative approach.

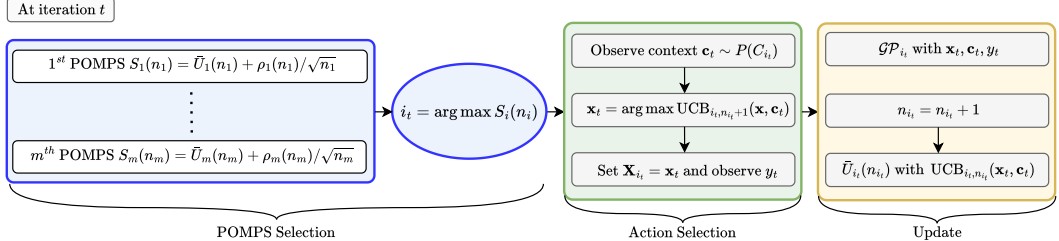

Figure 2: At each iteration $t$, we select a POMPS using the running average of the UCB evaluated at the points $\mathbf{x}_t$ and $\mathbf{c}_t$, plus an exploration term $\rho_i(n_i)/\sqrt{n_i}$, where $n_i$ is the number of times POMPS $i$ has been chosen so far. We then implement the intervention $\mathbf{X}_{i_t}$, observe the target $y_t$ under this intervention, and update the parameters of the algorithm. For more details please see Algorithm 2.

**High-level description**   Our method, outlined in Alg. 1, has 3 main components. The first component is the POMPS function, which computes the set $\mathbb{S}^* = \mathbb{S}^*[\mathcal{G}]$ (Lee & Bareinboim, 2020) for a given causal graph $\mathcal{G}$. This computation involves a brute-force search over MPSs, with computational complexity scaling exponentially in $|\mathbf{V}|$. However, this step is highly parallelisable, a feature we exploit in our implementation. Our definition of the contextual variables as $\mathbf{C}^* = \mathbf{V} \setminus (\mathbf{X}^* \cup \{Y\})$ suffices to ensure that optimal POMPSs are not missed (Lee & Bareinboim, 2020, Prop. 2). A visual overview is provided in Figure 2.

The method proceeds as follows: at each step $t$, based on historical data $\mathcal{D}^{t-1} := \{(\mathcal{S}_i, \pi_i, \mathbf{c}_i, y_i)\}_{i \in [t-1]}$, it first selects a POMPS $\mathcal{S}_t$ from $\mathbb{S}^*$, then chooses a policy $\pi_t \in \Pi_{\mathcal{S}_t}$.

---

**Algorithm 1** Contextual Causal BO (CoCa-BO)

---

**Input:** causal graph $\mathcal{G}$ on the nodes $\mathbf{V}$, the target $Y \in \mathbf{V}$, the set of intervenable variables $\mathbf{X}^* \subset \mathbf{V}$, domain of variables $\mathfrak{X}_\mathbf{V}$, number of iterations $T$. We consider $\mathbf{C}^* = \mathbf{V} \setminus (\mathbf{X}^* \cup \{Y\})$.
Compute $\mathbb{S}^* = \text{POMPS}(\mathcal{G}, \mathbf{X}^*)$ and create $A = \text{MAB}(\mathbb{S}^*)$
**for** $\mathcal{S} \in \mathbb{S}^*$ **do** Create $\text{Opt}_\mathcal{S} = \text{BO}(\mathbf{X}(\mathcal{S}), \mathbf{C}(\mathcal{S}), \mathfrak{X}_\mathcal{S})$
**for** $t = 1$ **to** $T$ **do**
   Select a POMPS: $\mathcal{S} = A.\text{suggest}()$
   Implement $\pi_\mathcal{S}$ provided by $\text{Opt}_\mathcal{S}$,
   Observe $y, \mathbf{x}_\mathcal{S}, \mathbf{c}_\mathcal{S}$ drawn from $P_\mathbf{V}^{\pi_\mathcal{S}}$
   $\text{Opt}_\mathcal{S}.\text{update}(y, \mathbf{x}_\mathcal{S}, \mathbf{c}_\mathcal{S}); A.\text{update}(y, \mathcal{S}, \text{Opt}_\mathcal{S})$
**end for**

---

This amounts to choosing an interventional value for $\mathbf{X}_{\mathcal{S}_t}$ after observing the context $\mathbf{c}_t$ associated with the variables $\mathbf{C}_{\mathcal{S}_t}$. Consequently, we may equivalently write the historical data as $\mathcal{D}^{t-1} = \{(\mathcal{S}_i, \mathbf{c}_i, \mathbf{x}_i, y_i)\}_{i \in [t-1]}$. The second component is the selection of a POMPS from the finite set $\mathbb{S}^*$ using MAB. Each POMPS serves as an arm, and the algorithm aims to find the one with the largest mean reward $\mu_\mathcal{S}^*$. Formally, we may view MAB as a mapping $A : \mathbb{S}^* \to \mathbb{R}$. At each step $t$, it selects a POMPS $\mathcal{S}_t \in \arg\max_{\mathcal{S} \in \mathbb{S}^*} A(\mathcal{S})$, and after observing $(\mathbf{c}_t, \mathbf{x}_t, y_t)$, it updates the value $A$ at $\mathcal{S}_t$ while keeping all others unchanged. Details of this update step are provided in Algorithm 2. The third component of our method is Bayesian Optimisation routine with Gaussian process surrogate, used to choose the intervention value of $\mathbf{X}_{\mathcal{S}_t}$ once the POMPS $\mathcal{S}_t$ is selected and the context $\mathbf{c}_t$ is observed. This component is described in more detail later in this section. Finally, the selected BO optimiser is updated at each iteration using the newly observed values. These updates involve Gaussian process posterior updates, which require inverting a $T \times T$ matrix, and thus have computational complexity $\mathcal{O}(T^3)$.

**Contextual Bayesian optimisation policy** For a given scope $\mathcal{S}$ selected at step $t$, the contextual Bayesian optimisation assumes that $y_t = g(\mathbf{x}_t, \mathbf{C}_t) + \sigma_n \xi_t$, where $g$ is a random Gaussian process defined on $\mathfrak{X}_{\mathbf{X}_\mathcal{S}} \times \mathfrak{X}_{\mathbf{C}_\mathcal{S}}$, $\mathbf{C}_t$ is a random vector drawn from $P_{\mathbf{C}_\mathcal{S}}^0$, and $\xi_t$ is a standard Gaussian random variable. All three quantities $(g, \mathbf{C}_t, \xi_t)$ are assumed independent. To provide more details, we define the set $[t]_\mathcal{S} = \{i \in [t] : \mathcal{S}_i = \mathcal{S}\}$ and denote by $\mathcal{D}_\mathcal{S}^t = \{(\mathbf{x}_\mathcal{S}^i, \mathbf{c}_\mathcal{S}^i, y^i)\}_{i \in [t]_\mathcal{S}}$ the subsample of $\mathcal{D}^t$ corresponding to those steps $i$ where the scope $\mathcal{S}$ was selected. The Gaussian process $g$ has zero mean and covariance function $k : (\mathfrak{X}_{\mathbf{X}_\mathcal{S}} \times \mathfrak{X}_{\mathbf{C}_\mathcal{S}})^2 \to \mathbb{R}$. Given the dataset $\mathcal{D}_\mathcal{S}^t$, the posterior distribution of the output for a test point $(\mathbf{x}, \mathbf{c})$ is a Gaussian $\mathcal{N}(\mu_{\text{post}}(\mathbf{x}, \mathbf{c}), \sigma_{\text{post}}^2(\mathbf{x}, \mathbf{c}))$, where

$$\mu_{\text{post}}(\mathbf{x}, \mathbf{c}) = \boldsymbol{k}_{[t]_\mathcal{S}}^\intercal(\mathbf{x}, \mathbf{c}) \left(\boldsymbol{K}_{[t]_\mathcal{S}} + \sigma_n^2 \mathbf{I}\right)^{-1} \mathbf{y}_{[t]_\mathcal{S}}$$

$$\sigma_{\text{post}}^2(\mathbf{x}, \mathbf{c}) = k((\mathbf{x}, \mathbf{c}), (\mathbf{x}, \mathbf{c})) - \boldsymbol{k}_{[t]_\mathcal{S}}^\intercal(\mathbf{x}, \mathbf{c}) \left(\boldsymbol{K}_{[t]_\mathcal{S}} + \sigma_n^2 \mathbf{I}\right)^{-1} \boldsymbol{k}_{[t]_\mathcal{S}}(\mathbf{x}, \mathbf{c}), \tag{1}$$

with $\mathbf{I}$ the identity matrix of appropriate size, $\mathbf{y}_{[t]_\mathcal{S}}$ the vector of observed targets in $\mathcal{D}_\mathcal{S}^t$, $\boldsymbol{K}_{[t]_\mathcal{S}}$ the kernel matrix of all $(\mathbf{x}, \mathbf{c})$ pairs in $\mathcal{D}_\mathcal{S}^t$, and $\boldsymbol{k}_{[t]_\mathcal{S}}(\mathbf{x}, \mathbf{c})$ the vector of covariances between the test point and each training point. The noise level $\sigma_n$ is a tunable parameter.

In our method, we deploy HEBO (Cowen-Rivers et al., 2022) for each POMPS, as it is a state-of-the-art BO algorithm (Turner et al., 2021) capable of optimising in contextual settings and handling both continuous and discrete variables. HEBO's Gaussian process model includes learnable transformations of both input and output, which makes it robust to heteroscedasticity and non-stationarity. All methods described in Section 6 use HEBO as the BO subroutine.

**POMPS selection** Our algorithm performs POMPS selection using historical values of the acquisition function combined with a margin (see Section B). In contrast, CaBO selects the next intervenable set based on the values of the acquisition functions associated with each GP. This strategy is motivated by the intuition that the optimal intervenable set, together with its optimal interventional values, should yield the highest reward. Accordingly, acquisition functions are expected to reflect this pattern, particularly in regions of low uncertainty. However, this approach becomes unsuitable in contextual settings, where the optimal values of acquisition functions inherently depend on the observed context. This dependence introduces a fundamental issue: the comparison is performed between acquisition functions evaluated at different contexts, which undermines the validity of such comparisons. Such

comparisons may lead the optimisation process to switch arbitrarily between scopes based on the observed context, potentially causing instability or divergence.

Our selection criterion $\bar{U}_{i_t}$, defined in Algorithm 2, is the running average of the UCB values over the evaluation points for each POMPS instance. As shown in Lemma 2, this quantity simultaneously upper and lower bounds the empirical mean of the target under the optimal policy, up to additive terms. These terms decay sufficiently fast to ensure that suboptimal POMPS are selected only rarely, as formalized in Lemma 4.

## 4    REGRET BOUND FOR COCA-BO

In this section, we prove that the CoCa-BO algorithm outlined in Algorithm 2 achieves sublinear regret. For this purpose, we place ourselves in a more general setting, which we refer to as multi-function BO. We assume that we are given an integer $m \in \mathbb{N}^*$, together with two sequences of sets $(\mathfrak{X}_{\mathbf{X}(i)})_{i \in [m]}$ and $(\mathfrak{X}_{\mathbf{C}(i)})_{i \in [m]}$. At each step $t \in \mathbb{N}^*$, we first select an index $i_t \in [m]$ and then observe a random variable $\mathbf{C}_t$ drawn from $\mathfrak{X}_{\mathbf{C}(i_t)}$. Based on $\mathbf{C}_t$ and the past observations, we choose $\mathbf{X}_t$ from $\mathfrak{X}_{\mathbf{X}(i_t)}$ and subsequently observe the reward $Y_t$. In this formulation, the indices $i \in [m]$, henceforth called *arms*, correspond to POMPSs; the variables $\mathbf{C}_t$ correspond to *contexts*; and the variables $\mathbf{X}_t$, henceforth called *actions*, correspond to interventional values.

We assume that for some random processes $f_1, \ldots, f_m$ and for some unknown context distributions $P_1, \ldots, P_m$, the reward takes the form $Y_{i_t} = f_{i_t}(\mathbf{X}_t, \mathbf{C}_t) + \epsilon_t$, where conditionally to $i_t, f_1, \ldots, f_m$, $\mathbf{C}_t$ is drawn from $P_{i_t}$, and conditionally to $i_t, f_1, \ldots, f_m$ and $\mathbf{C}_t$, $\epsilon_t \sim \mathcal{N}(0, \sigma^2)$. A more detailed description of the setting can be found in Section B.1.

**Assumption 1** (Convex-compact domains). For every $i \in [m]$, $\mathfrak{X}_{\mathbf{X}(i)}$ and $\mathfrak{X}_{\mathbf{C}(i)}$ are convex and compact. Furthermore, there exist constants $r_i, r_i' > 0$, $d_i, d_i' \in \mathbb{N}$ such that $\mathfrak{X}_{\mathbf{X}(i)} \subset [0, r_i]^{d_i}$ and $\mathfrak{X}_{\mathbf{C}(i)} \subset [0, r_i']^{d_i'}$ for every $i \in [m]$. We set $\bar{r}_i = r_i \vee r_i'$ and $\bar{d}_i = d_i \vee d_i'$.

**Assumption 2** (Smooth and bounded kernels). Processes $f_i$ are independent, each $f_i$ be Gaussian with zero mean and covariance kernel $k_i$. The covariance kernels $k_i$ are bounded, *i.e.*, for some $\kappa_1, \ldots, \kappa_m > 0$, we have $\max_{\mathbf{x}, \mathbf{c}} k_i\big((\mathbf{x}, \mathbf{c}); (\mathbf{x}, \mathbf{c})\big) \leq \kappa_i$. Furthermore, there are constants $\psi, \varphi > 0$ such that, for any $\delta \in (0, 1)$ and for any $i \in [m]$, with probability at least $1 - \delta$

$$\sup_{\mathbf{x}, \mathbf{c}} \|\nabla f_i(\mathbf{x}, \mathbf{c})\|_\infty \leq \varphi \sqrt{\log(\bar{d}_i \psi / \delta)}. \tag{2}$$

Note that these assumptions are not very restrictive compared to those usually imposed in Bayesian optimisation. In particular, it follows from (Ghosal & Roy, 2006) that if the kernels $k_i$ admit fourth-order mixed partial derivatives, then the path smoothness condition (2) is satisfied.

For brevity, we defer the precise definitions of the subroutines MAB-UCB and HEBO from Algorithm 1 to Section B. We only mention here that, in the Bayesian optimisation subroutine, an upper confidence bound is constructed from the posterior means $\mu_{t-1}^{(i)}$ and standard deviations $\sigma_{t-1}^{(i)}$ of the $i$-th GP at time $t$, together with a sequence of tuning parameters $(\beta_i(n))_{i \in [m], n \in \mathbb{N}^*}$. As for the arm-selection subroutine MAB-UCB, it chooses the arm with the largest average UCB over past observations, plus a penalty term involving the sequence of tuning parameters $(\rho_i(n))_{i \in [m], n \in \mathbb{N}^*}$. To ease that statement of the main theorem, we set

$$\beta_T = \max_{i \in [m]} \max_{n \leq T} \beta_i(n) \qquad \text{and} \qquad \rho_T = \max_{i \in [m]} \max_{n \leq T} \rho_i(n).$$

Let $\mu_i = \mathbb{E}[f_i^*(\mathbf{C}_{i,1}) \mid f_i] = \int \max_{\mathbf{x}} f_i(\mathbf{x}, \mathbf{c}) \, P_i(\mathrm{d}\mathbf{c})$ and $I^* = \operatorname{argmax}_{i \in [m]} \mu_i$. Clearly, $\mu_i$ depends on $f_i$ and $I^*$ depends on $\boldsymbol{f} = (f_1, \ldots, f_m)$, but for simplicity of notation we do not make this dependence explicit. We also write $\mu^* = \max_{i \in [m]} \mu_i$. The cumulative regret is then

$$R_T = \sum_{t=1}^T \big\{ \mu^* - \mathbb{E}\big[ f_{i_t}(\mathbf{X}_t, \mathbf{C}_t) \mid \boldsymbol{f}, \mathcal{D}^{t-1} \big] \big\}.$$

In words, the regret is the difference between the context-averaged reward of an oracle strategy (which always selects an arm with maximal context-averaged reward and plays a context-optimal action) and the context-averaged reward of our algorithm. This notion of regret, specific to multi-function

BO, differs from those considered in the MAB literature (Lattimore & Szepesvári, 2020) and the BO literature (Srinivas et al., 2010). In MAB, the reward of a chosen arm is constant across rounds, whereas in our setting, the expected reward evolves as the estimated maxima of $f_{i_t}$ are progressively refined. On the other hand, unlike in BO, our regret compares the context-averaged maximum of $f_{i_t}$ to $\mu^*$, which may correspond to the optimal value of another function $f_j$. These distinctions introduce additional challenges in establishing a bound for regret $R_T$ of the procedure defined in Algorithm 1.

A key quantity characterising the regret of the algorithms commonly used in BO with Gaussian process prior is the maximum information gain (Srinivas et al., 2010). Denoted by $\gamma_i(n)$, the maximum information gain of the $i^{th}$ GP $f_i$ over horizon $n$, is defined by

$$\gamma_i(n) = \max_{A:|A|=n} \frac{1}{2} \log \det(\mathbf{I} + \sigma^{-2} \boldsymbol{K}_{i,A}),$$

where the maximum is over all sets $A \subset \mathfrak{X}_{\mathbf{X}(i)} \times \mathfrak{X}_{\mathbf{C}(i)}$ of cardinality $n$ and $\boldsymbol{K}_{i,A}$ is the $n \times n$ matrix with the general term $K_i(\mathbf{z}, \mathbf{z}')$, $\mathbf{z}, \mathbf{z}' \in \mathfrak{X}_{\mathbf{X}(i)} \times \mathfrak{X}_{\mathbf{C}(i)}$. We refer the reader to (Vakili et al., 2021) for results on bounding the maximum information gain in terms of $n$ and the kernel parameters. Based on $\gamma_i(n)$, we define the worst-case-arm and arm-averaged information gains by

$$\gamma_T = \max_{i \in [m]} \gamma_i(T), \qquad \text{and} \qquad \bar{\gamma}_T = \max_{\mathbf{n}:\sum_i n_i = T} \frac{1}{m} \sum_{i=1}^{m} \gamma_i(n_i),$$

where the last maximum is over all vectors $\mathbf{n} = (n_1, \ldots, n_m) \in \mathbb{N}^m$ summing to $T$.

**Theorem 1.** *Let $\delta \in (0, 1/2)$, $\boldsymbol{\Delta} = (\Delta_1, \ldots, \Delta_m)$ be the vector of sub-optimality gaps defined by $\Delta_i = \mu^* - \mu_i$ and $I^* = \{i : \Delta_i = 0\}$. Let Assumptions 1 and 2 be fulfilled. There exist constants $\mathsf{A}_1$, $\mathsf{A}_2$, $\mathsf{A}_3$ depending only on $\sigma$, $\varphi$, $\psi$, $(\kappa_i, r_i, r_i')_{i \in [m]}$, such that if the parameters $\beta_i$ and $\rho_i$ satisfy*

$$\beta_i(n) \wedge \rho_i(n) \geq \mathsf{A}_1 \bar{d}_i \log\left(\mathsf{A}_1 m \bar{d}_i n/\delta\right)$$

*for every $i \in [m]$ and every $n \in \mathbb{N}^*$, then regret (6) of UCB-BO Algorithm 2 applied up to horizon $T$, with probability at least $1 - 2\delta$, satisfies for all $T \in \mathbb{N}^*$ the following inequalities*

$$R_T \leq \mathsf{A}_2\left\{\sqrt{mT}\left(\sqrt{\beta_T \bar{\gamma}_T} + \rho_T\right)\right\} + \|\boldsymbol{\Delta}\|_1, \tag{3}$$

$$R_T \leq \mathsf{A}_3\left\{\sqrt{|I^*|T}\left(\sqrt{\beta_T \bar{\gamma}_T} + \rho_T\right) + \left(\beta_T \gamma_T + \rho_T^2\right) \sum_{i \notin I^*} \frac{1}{\Delta_i} + m\left(\sqrt{\beta_1} + \rho_T\right)\right\} + \|\boldsymbol{\Delta}\|_1. \tag{4}$$

We now provide several comments on the upper bounds in Theorem 1. First, the term $\|\boldsymbol{\Delta}\|_1$ in the regret bounds is unavoidable. It arises because the algorithm plays each of the $m$ arms independently during the initial $m$ rounds. However, since this term is $O(m)$ and independent of $T$, its impact on the asymptotic regret is negligible. Aside from this $O(m)$ term, the bound in (3) is independent of the functions $(f_1, \ldots, f_m)$; we therefore refer to it as a worst-case bound. For kernels with exponential eigendecay (e.g., the squared exponential kernel), the parameter $\gamma_T$ is at most polylogarithmic in $T$ (see Table 3 in the supplementary material). Hence, with appropriate tuning, the worst-case regret is of order $\sqrt{mT} \max_i \bar{d}_i$, up to polylogarithmic factors in $m$ and $T$. Notably, this yields a $\sqrt{T}\operatorname{polylog}(T)$ dependence on $T$, a significant improvement over a linear regret.

One might expect the regret dependence on $m$ to improve beyond $\sqrt{m}$ when there is a significant gap between the optimal and sub-optimal arms. This is indeed the case, as captured by the instance-dependent bound in (4). Under mild assumptions on the Gaussian processes (see Section C.2), the set of optimal arms $I^*$ is almost surely a singleton. In such cases, the leading term of (4) is of order $\sqrt{T} \max_i \bar{d}_i$ (up to polylog factors), improving upon the worst-case bound by a factor of $\sqrt{m}$. This improvement holds when the sub-optimality gaps $\Delta_i$ for $i \notin I^*$ are bounded away from zero; otherwise, the term $\sum_{i \notin I^*} 1/\Delta_i$ may dominate.

To the best of our knowledge, Theorem 1 presents the first regret bound not only for our proposed contextual-causal BO framework, but also for the more established Causal BO (CaBO) setting.

## 5 RELATED WORK

The integration of causal knowledge to improve policy learning is an active area of research (Lee & Bareinboim, 2018; 2019; 2020), with recent progress on both conditional and unconditional

policies (Aglietti et al., 2020b; Zhang & Bareinboim, 2017; 2022). Our work builds on and extends this literature. Below, we summarize the most related lines of work and contrast them with our contributions.

Causal Bayesian Optimisation (Aglietti et al., 2020b) uses the causal graph and a causal acquisition function (cAF) to search over policy scopes, treating non-intervenable variables as unobserved. In contrast, our method explicitly leverages contextual variables and searches the joint space of intervenable and contextual variables. This not only leads to better solutions but also overcomes the instabilities of cAF in contextual scenarios. Contextual Bayesian Optimisation conditions interventional choices on observed contexts but typically treats all controllable variables as intervenable. This can prevent recovery of the optimal policy. By searching over the set of POMPSs, our method avoids this suboptimality.

Functional Causal BO (Gultchin et al., 2023) places a GP prior on $\mu(\pi)$ via a finite RKHS expansion, $\pi(\cdot) = \sum_{i=1}^{N} \alpha_i k'_\mathcal{S}(\mathbf{c}^i, \cdot)$. Although this permits distance-based kernels, the critical choices of $N$, $\alpha_i$, and $\mathbf{c}^i$ are unspecified, hindering reproducibility. Furthermore, the optimisation of acquisition functions within this framework is also not addressed. In contrast, our method requires no such finite expansion, avoids the associated optimisation overhead, provides theoretical guarantees, and is accompanied by a public implementation tested on diverse environments.

Model-Based Causal BO (Sussex et al., 2023) assumes specific structural causal models ($V = f_V(\mathbf{PA}_V) + U_V$) with restrictions on the noise terms and requires no unobserved confounders. Without any reduction of the search space, it considers only a limited family of interventions representable as parameter perturbations of the structural functions $f_V$. Sussex et al. (2023) prove that $R_T \leq \mathcal{O}\big(|\mathbf{V}| (K\beta_T)^N \sqrt{T\gamma_T}\big)$, where $K$ is the maximum in-degree, $N$ is the length of the longest path to $Y$, $\beta_T$ and $\gamma_T$ are similar to those defined in Section 4. Note that both the factor $(K\beta_T)^N$ and $|\mathbf{V}|$ can be very large (see Section A.3). In contrast, our theory applies under more general conditions and the bounds scale more favorably; in the setting with no unobserved confounders, the set of POMPS is a singleton (Lee & Bareinboim, 2020), leading to significantly improved scaling.

The method of (Zhang & Bareinboim, 2022) also optimises over POMPS. However, it restrictively assumes that the domain $\mathfrak{X}_\mathbf{V}$ is discrete and finite, an assumption not required by our approach. The regret bound established for causal-UCB$^*$ is sublinear in $T$, but it scales at least as the sum of the square roots of the cardinalities of the domains of the variables in $\mathbf{V}$.

Another MAB-based method that utilises causal knowledge is proposed by Lattimore et al. (2016). It is the first to show that a causal bandit framework subsumes the classical setting. Their method enjoys favourable regret bounds and is computationally efficient. As with Zhang & Bareinboim (2022), it assumes all variables have finite support. However, it does not consider contextual interventions and assumes the absence of unobserved confounding.

Multitask GPs (Bonilla et al. (2007)) learn similarities across the outputs of multiple functions and are useful for optimisation over multiple tasks that share inputs. Aglietti et al. (2020a) formalise when information sharing is possible under a known causal graph: tasks differ in their sets of interventional variables but share the same target variable. They show that sharing is possible only if every variable confounded with the target does not simultaneously have unconfounded incoming and outgoing edges. However, they study optimisation without contextual variables, and their information sharing mechanism requires computing high-dimensional integrals. Identifying assumptions that enable information sharing between POMPS without strong restrictions, and a computationally efficient mechanism to realise it, remains an open problem.

## 6 EXPERIMENTS

We evaluate the proposed method against CaBO and CoBO across a series of experiments. For each experiment and optimiser, we report the time-normalised cumulative regret $\bar{R}_T = \frac{1}{T} \sum_{t=1}^{T} \hat{r}_t$ where $\hat{r}_t = \mathbb{E}_{\pi^*}[Y] - y_t$ is the immediate pseudo-regret at iteration $t$. We also track the cumulative MPS selection frequency, i.e., the fraction of times each MPS has been selected up to iteration $t$. We consider two configurations.

**Configuration I:** Both the proposed method and the benchmark method can achieve the optimum. We aim to assess the additional cost introduced by the proposed method in terms of sample com-

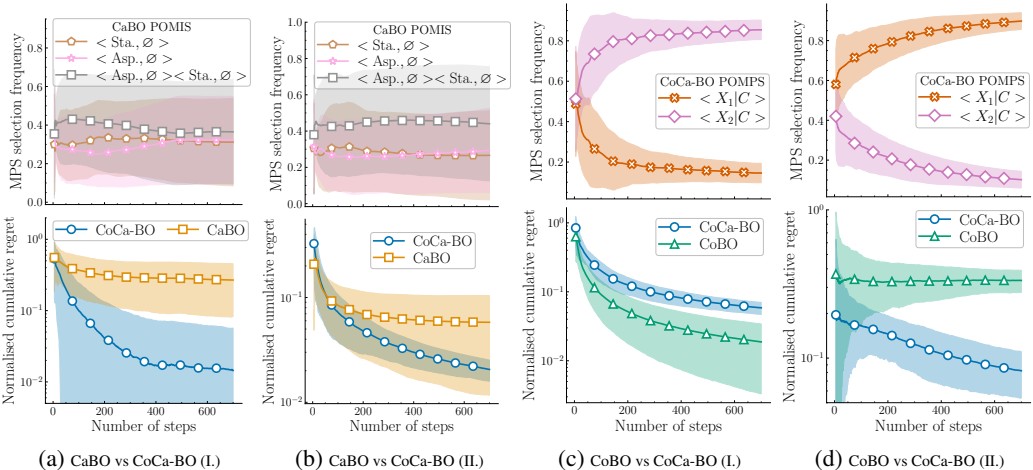

Figure 3: Top: frequency of selecting the corresponding POMPS or POMIS (omitted when there is only one candidate). Bottom: time-normalised cumulative regret $\bar{R}_T$.

plexity. In environments with many observed variables, CoBO results in an expansive optimisation domain, requiring a substantial sample size. Our method leverages the independencies encoded within the causal graph to reduce the optimisation domain. For instance, in another example reported in Appendix A.3, the dimensionality of the domain is reduced by a factor of 22.

**Configuration II:** The benchmark method cannot achieve the optimum. We empirically confirm that there are cases where neither CoBO nor CaBO converges to the optimum, leading to linear cumulative regret. In the same cases, we demonstrate that CoCa-BO converges to the optimum policy and has sub-linear cumulative regret.

All experiments were conducted with 110 random seeds over 700 iterations, corresponding to 700 distinct interventions. We used a single-node system with 120 CPUs, though the experiments can also be reproduced on a single-core machine with 16 GB RAM. Results are reported as averages, with error bars denoting 1.96 standard errors.

**CaBO vs CoCa-BO** To compare CaBO and CoCa-BO under the two aforementioned configurations, we consider two SCMs based on the graph of Fig. 4 below (Ferro et al., 2015; Thompson, 2019).

In this example, the variables PSA, age, BMI, cancer, aspirin and statin are observable, but only the last two are intervenable. Prostate-specific antigen (PSA) (Wang et al., 2011), used to detect prostate cancer, is our target. Here, the set of POMISs is $\{\varnothing, \{Aspirin\}, \{Statin\}, \{Aspirin, Statin\}\}$ and there is only one POMPS, namely $\{(Aspirin; Age, BMI); (Statin; Age, BMI)\}$.

*First Configuration*: When SCM is such that the optimal value of the target is the same across all values of the contextual variables, CaBO can attain the optimum. This requires specific alignment of the SCM parameters, which may be unstable and sensitive to small fluctuations (Pearl, 2009, Ch. 2). Such an SCM is defined in Section A.1.1. The optimal values at which Statin and Aspirin are controlled to minimise PSA are the same for all values of Age and BMI, that is $\forall(\text{age}, \text{bmi}) \in \mathfrak{X}_{\text{Age}} \times \mathfrak{X}_{\text{BMI}}$:

$$\underset{(\text{aspirin}, \text{statin})}{\arg\min} \mathbb{E}[Y \mid \text{do}(\text{aspirin}, \text{statin}), \text{age}, \text{bmi}] = \underset{(\text{aspirin}, \text{statin})}{\arg\min} \mathbb{E}[Y \mid \text{do}(\text{aspirin}, \text{statin})].$$

We show that the minimum above is attained for aspirin$= 0$ and statin$= 1$. Empirical results for this SCM appear in the first column of Fig. 3a. Since CaBO selects POMISs via the causal acquisition function, the top plot shows that its choices are highly variable, which harms optimiser convergence (bottom plot). In contrast, CoCa-BO achieves much smaller regret, as illustrated in the bottom plot.

*Second Configuration*: The SCM for the second configuration, defined in Appendix A.1.2, is consistent with the causal graph in Fig. 4. It implies that controlling Aspirin and Statin based on Age and BMI yields a lower expected PSA than fixing them to constant values. Fig. 3b depicts the results for this setup. Since CaBO marginalises the contextual variables, it cannot control medications based on Age and BMI, and therefore fails to converge (bottom plot). The top plot shows again the high variability in POMIS selection under CaBO. As expected, CoCa-BO attains a much smaller regret.

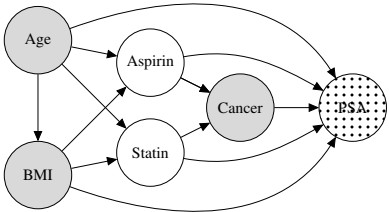

Figure 4: Causal graph of PSA level. White nodes: intervenable variables; gray nodes: observable variables; shaded node: target variable.

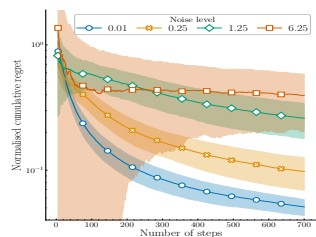

Figure 5: Robustness to the noise: Normalised regret under varying noise levels.

**CoBO vs CoCa-BO**  We consider two SCMs based on the graph of Fig. 1.

*First Configuration:* Appendix A.2 describes an SCM consistent with the causal graph in Fig. 1, where intervening on both $X_1$ and $X_2$ does not hinder the agent's ability to reach the optimal outcome. Fig. 3c (bottom plot) shows both methods converge to the optimum, as evidenced by the decreasing normalised cumulative regret, with CoBO converging faster than CoCa-BO. In this configuration, CoCa-BO's advantage over CoBO emerges in environments with many variables, as shown in Section A.3.

*Second Configuration:* The second configuration examined in this study is the SCM of Fig. 1. We have shown analytically that the CoBO method is unable to attain the optimal solution. Our method, CoCa-BO, has been demonstrated to effectively achieve the desired optimum, resulting in sub-linear regret. The top sub-figure of Fig. 3d illustrates the cumulative frequency of selecting the corresponding POMPS. We observe that CoCa-BO converges to the policy scope $(X_1; C)$ yielding the optimal value. The lower sub-figure of Fig. 3d illustrates that the normalised cumulative regret remains constant for CoBO, indicating its linear cumulative regret. On the other hand, CoCa-BO attains sub-linear cumulative regret, as evidenced by the decreasing trend of the blue curve.

**Robustness to Noise**  The target variable $Y$ is pivotal, as its observed values guide both policy scope and selection. Consequently, higher noise in $Y$ slows the convergence of optimisation algorithms (Lattimore & Szepesvári, 2020; Krause & Ong, 2011; Srinivas et al., 2010). To assess the robustness of CoCa-BO to noise in $Y$, we vary the standard deviation of $\epsilon_Y$ in the example of Appendix A.2, adjusting $\sigma(\epsilon_Y)$ from 0.01 to 6.25. As shown in Fig. 5, CoCa-BO converges for $\sigma$ up to 1.25, while convergence becomes unclear under extreme noise, e.g., $\sigma(\epsilon_Y) = 6.25$.

## 7    CONCLUSION

We provided evidence that existing Bayesian optimisation methods that incorporate causal knowledge are not capable of integrating contextual information without incurring substantial computational and statistical costs. To overcome these challenges, we introduced CoCa-BO, a novel method that efficiently optimises over mixed policy scopes. We derived worst-case and instance-dependent regret bounds, which also address theoretical gaps in previous methods. We conducted experiments across various environments and demonstrated that CoCa-BO consistently reached the optimum even where existing methods could not. Moreover, our experiments showed that CoCa-BO is robust to noise and reduces sample complexity by identifying smaller optimisation domains in environments with a large number of variables. To support reproducibility, we provide our Pyro-based implementation (Bingham et al., 2018; Phan et al., 2019), offering a unified framework for future research.

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

# Appendix

## Table of Contents

| Acronym | Full text |
|---------|-----------|
| BO | Bayesian Optimisation |
| CaBO | Causal Bayesian Optimisation |
| CoBO | Contextual Bayesian Optimisation |
| CoCa-BO | Contextual Causal Bayesian Optimisation |
| POMIS | Possibly Optimal Minimal Intervention Set |
| SCM | Structural Causal Model |
| GP | Gaussian Process |
| UCB | Upper Confidence Bound |
| DAG | Directed Acyclic Graph |
| MPS | Mixed Policy Scope |
| POMPS | Possibly-Optimal Mixed Policy Scope |
| MAB | Multi-Armed Bandit |

Table 1: This table presents the main acronyms used throughout the paper.

# A    ADDITIONAL DETAILS ON THE EXPERIMENTAL RESULTS

In this section, we present the details necessary to understand and reproduce the experiments reported in Section 6.

## A.1    CABO VS COCA-BO

We describe and analyse SCMs used for comparing CaBO with CoCa-BO in this section.

### A.1.1    FIRST CONFIGURATION

The SCM that is used for the comparison of CaBO vs CoCa-BO in a scenario where CaBO can achieve the optimum is given below: Age $\sim \mathcal{U}(55, 75)$, $\epsilon \sim \mathcal{N}(0, 0.4)$ and[2]

$$
\begin{aligned}
\text{BMI} &\sim \mathcal{N}(27.0 - 0.01 \times \text{Age}, 0.7) \\
\text{Aspirin} &= \varrho(-8.0 + 0.10 \times \text{Age} + 0.03 \times \text{BMI}) \\
\text{Statin} &= \varrho(-13.0 + 0.10 \times \text{Age} + 0.20 \times \text{BMI}) \\
\text{Cancer} &= \varrho(2.2 - 0.05 \times \text{Age} + 0.01 \times \text{BMI} - 0.04 \times \text{Statin} + 0.02 \times \text{Aspirin}) \\
\text{PSA} &= \epsilon + 6.8 + 0.04 \times \text{Age} - 0.15 \times \text{BMI} - 0.60 \times \text{Statin} + 0.55 \times \text{Aspirin} + \text{Cancer}
\end{aligned}
$$

We first note that for any values of Age and BMI

$$
\begin{aligned}
\mathbb{E}[\text{PSA} \mid &\operatorname{do}(\text{asp}, \text{sta}), \text{age}, \text{bmi}] \\
&= \mathbb{E}[\epsilon + 6.8 + 0.04\,\text{Age} - 0.15\text{BMI} - 0.60\,\text{Sta} + 0.55\,\text{Asp} + \text{Can} \mid \operatorname{do}(\text{asp}, \text{sta}), \text{age}, \text{bmi}] \\
&= 6.8 + 0.04\text{age} - 0.15\text{bmi} - 0.6\text{sta} + 0.55\text{asp} + \mathbb{E}[\text{Can} \mid \operatorname{do}(\text{asp}, \text{sta}), \text{age}, \text{bmi}] \\
&= 6.8 + 0.04\text{age} - 0.15\text{bmi} - 0.6\text{sta} + 0.55\text{asp} \\
&\quad + \varrho(2.2 - 0.05\text{age} + 0.01\text{bmi} - 0.04\text{sta} + 0.02\text{asp})
\end{aligned}
$$

It is easy to see that

$$
\begin{aligned}
\operatorname*{argmin}_{\text{asp}, \text{sta}} \mathbb{E}[\text{PSA} \mid &\operatorname{do}(\text{aspirin}, \text{statin}), \text{age}, \text{bmi}] = \\
&\operatorname*{argmin}_{\text{asp}, \text{sta}} -0.6\,\text{sta} + 0.55\,\text{asp} + \varrho(2.2 - 0.05\text{age} + 0.01\text{bmi} - 0.04\text{sta} + 0.02\text{asp}).
\end{aligned}
$$

In summary, to minimise PSA, Statin should be controlled at its highest value and Aspirin at its lowest value. Since the domains of both variables are $[0, 1]$, the solution of the optimisation problem is Aspirin $= 0$ and Statin $= 1$, which is independent of Age and BMI. This implies that the optimal control values for Aspirin and Statin are the same for all values of Age and BMI. This is why the optimum is achievable by CaBO.

---

[2]Here, $\varrho$ denotes the sigmoid function.

| Notation | Meaning |
|---|---|
| $\mathbf{U}, \mathbf{V}, \mathbf{X}$ | Vectors/sets composed of random variables |
| $\mathbf{u}, \mathbf{v}, \mathbf{x}$ | Values taken by vectors/sets of random variables |
| $U, V, X$ | random variables |
| $\mathfrak{X}_U, \mathfrak{X}_V, \mathfrak{X}_X$ | The domains of the random variables $U, V, X$; *i.e.*, the sets in which $U, V, X$ take their values. |
| $\mathfrak{X}_{\mathbf{U}}, \mathfrak{X}_{\mathbf{V}}, \mathfrak{X}_{\mathbf{X}}$ | The domains of the random vectors $\mathbf{U}, \mathbf{V}, \mathbf{X}$. |
| $P_{\mathbf{V}}^{\square}$ | A probability distribution of the random vector $\mathbf{V}$; $\square$ can be replaced by different symbols, for instance, by 0 or by $\pi$. |
| $\mathbb{E}_{P_{\mathbf{V}}^{\square}}[Z]$ | The expectation of the random variable $Z = g(\mathbf{V})$ assuming that $\mathbf{V}$ is drawn from $P_{\mathbf{V}}^{\square}$. |
| $\mathcal{F}$ | Set of functions with elements usually denoted by $f$. |
| $SCM$ | A structural causal model given by the quadruple $(\mathbf{V}, \mathbf{U}, \mathcal{F}, P_{\mathbf{U}})$. |
| $\mathcal{G}$ | A directed acyclic graph. |
| $\mathbf{PA}_V$ | When $V$ is a node in a DAG, $\mathbf{PA}_V$ is the set of its parent nodes. |
| $\mathbf{C}_X$ | When $X$ is an intervenable variable, $\mathbf{C}_X$ is its scope, *i.e.*, a set of observable variables that can be used for planning an intervention on $X$. |
| $\mathcal{S}$ | A mixed policy scope, *i.e.*, a collection of pairs $(X, \mathbf{C}_X)$. |
| $\mathbf{X}(\mathcal{S})$ | The set of intervenable variables of $\mathcal{S}$. |
| $\mathbf{C}(\mathcal{S})$ | The set of contextual variables of $\mathcal{S}$. |
| $\mathcal{G}_{\mathcal{S}}$ | The graph obtained from $\mathcal{G}$ by removing edges into $X$ and adding new ones from each element of $\mathbf{C}_X$ to $X$ for every $(X, \mathbf{C}_X) \in \mathcal{S}$. |
| $\pi_{X \mid \mathbf{C}_X}$ | a mapping from $\mathfrak{X}_{\mathbf{C}_X}$ to $\mathfrak{X}_X$. |
| $\mathbb{S}[\mathcal{G}]$ | All mixed policy scopes $\mathcal{S}$ compatible with the DAG $\mathcal{G}$. |
| $\pi$ | A mixed policy, *i.e.*, a collection of mappings $\pi_{X \mid \mathbf{C}_X}$ |
| $\Pi_{\mathcal{S}}$ | The set of all mixed policies $\pi$ based on the mixed policy scope $\mathcal{S}$. |
| $\Pi_{\mathcal{G}}$ | The set of all mixed policies $\pi$ based on a mixed policy scope $\mathcal{S} \in \mathbb{S}[\mathcal{G}]$. |
| $\mathfrak{M}^{\pi}$ | The SCM obtained from $\mathfrak{M}$ by applying the policy $\pi$. |
| $\mu(\pi)$ | The expected reward under the mixed policy $\pi$. For a well specified $Y \in \mathbf{V}$, it is defined as $\mu(\pi) = \mathbb{E}_{P_{\mathbf{V}}^{\pi}}[Y]$. |
| $\mu_{\mathcal{S}}^*$ | The highest expected reward among all mixed policies $\pi$ based on the scope $\mathcal{S}$. |
| $\mathbb{E}_{\pi}[Z]$ | For a random variable $Z = g(\mathbf{V})$ and a mixed policy $\pi$, this is a shorthand for $\mathbb{E}_{P_{\mathbf{V}}^{\pi}}[Z]$. |
| $\mathbb{S}^*[\mathcal{G}]$ | The subset of $\mathbb{S}[\mathcal{G}]$ consisting of only those $\mathcal{S}$ that are possibly optimal. |
| $\mathcal{D}^t$ | Historical data up to time $t \in \mathbb{N}$, consisting of quadruplets $(\mathcal{S}_i, \mathbf{c}_i, \mathbf{x}_i, y_i)$ for $i = 1, \ldots, t$. |
| $\mu_t^{(i)}, \sigma_t^{(i)}$ | Posterior mean and standard deviation of the $i^{th}$ Gaussian process after time $t$. |
| $\hat{r}_t, R_t, \bar{R}_t$ | Instantaneous regret at time $t$, cumulative regret at time $t$, and time-normalised cumulative regret at time $t$. |

Table 2: This table summarises the main notations used throughout the paper.

### A.1.2 SECOND CONFIGURATION

The SCM that is used for this setup is such that an agent should control Aspirin and Statin based on Age and BMI to achieve the minimum expected value for PSA. It is defined by Age $\sim \mathcal{U}(55, 75)$,

$\epsilon \sim \mathcal{N}(0, 0.01)$ and

$$
\begin{aligned}
\text{BMI} &\sim \mathcal{N}(27.0 - 0.01 \times \text{Age}, 0.1) \\
\text{Aspirin} &= \varrho(-8.0 + 0.10 \times \text{Age} + 0.03 \times \text{BMI}) \\
\text{Statin} &= \varrho(-13.0 + 0.10 \times \text{Age} + 0.20 \times \text{BMI}) \\
\text{Cancer} &= \text{Statin}^2 + \text{AgeBMI}^2 + 0.5 \times \text{Aspirin}^2 \\
\text{PSA} &= \epsilon + \text{Cancer} + 0.5 \times \text{Asp}^2 + \text{AgeBMI}^2 - 2 \times \text{AgeBMI} \times (\text{Asp} + \text{Sta}),
\end{aligned}
$$

where we used the notation

$$
\text{AgeBMI} = \Big(\frac{\text{Age} - 55}{21}\Big)\Big|\frac{\text{BMI} - 27}{4}\Big|.
$$

Plugging in the structural equation of Cancer into PSA, we get

$$
\begin{aligned}
\text{PSA} &= \epsilon + \text{Asp}^2 + \text{Sta}^2 + 2\text{AgeBMI}^2 - 2 \times \text{AspAgeBMI} - 2 \times \text{Sta} \times \text{AgeBMI} \\
&= (\text{Asp} - \text{AgeBMI})^2 + (\text{Sta} - \text{AgeBMI})^2 + \epsilon
\end{aligned}
$$

An agent controlling both Aspirin and Statin at $\Big(\frac{\text{Age}-55}{21}\Big)\Big|\frac{\text{BMI}-27}{4}\Big|$ level achieves the minimal expected value of PSA, namely 0. CaBO fails as it does not consider Age and BMI while controlling Aspirin and Statin.

## A.2 COBO VS COCA-BO

We introduce an SCM such that CoBO which by default controls both $X_1$ and $X_2$ based on $C$ is still capable of getting to the optimum value. The SCM is based on the causal graph depicted in Fig. 1 and the following equations

$$
\begin{aligned}
U_1, U_2 &\sim \text{Uniform}(-1, 1) \\
\epsilon_C, \epsilon_{X_1}, &\sim \mathcal{N}(0, 0.1), \ \epsilon_Y \sim \mathcal{N}(0, 0.01) \\
C &= U_1 + \epsilon_C \\
X_1 &= U_1 + \epsilon_{X_1} \\
X_2 &= |C - X_1| + 0.2U_2 \\
Y &= \cos(C - X_2) + 0.1U_2 + \epsilon_Y.
\end{aligned}
$$

This SCM has two POMPSs, namely $(X_1; C)$ and $(X_2; C)$.

We first note that intervening on both $X_1$ and $X_2$ is redundant in the sense that intervening on $X_2$ destroys the directed causal path from $X_1$ to $Y$ and renders the intervention on $X_1$ irrelevant to $Y$. Let the intervention on $X_2$ be given by an arbitrary function $\pi : \mathfrak{X}_C \mapsto \mathfrak{X}_{X_2}$. Then the expectation of $Y$ under such control can be expressed as:

$$
\mathbb{E}_\pi[Y] = \mathbb{E}_\pi[\cos(C - X_2) + 0.1U_2 + \epsilon_y] = \mathbb{E}_\pi[\cos(C - X_2)] = \mathbb{E}_\pi[\cos(C - \pi(C))].
$$

It is trivial to see that to maximise the above expectation, one should set $X_2$ at the value of $C$, *i.e.*, $\pi$ is the identity. This yields $\max_\pi \mathbb{E}_\pi[\cos(C - \pi(C))] = 1$.

Therefore intervening on $X_2$ based on $C$ yields the optimal value, as 1 which is the highest value that the expectation of $Y$ may achieve under any policy $\mathbb{E}_\pi[Y] = \mathbb{E}_\pi[\cos(C - X_2) + 0.1U_2 + 0.1\epsilon_y] = \mathbb{E}_\pi[\cos(C - X_2)] \leq 1$. We conclude that CoBO can achieve the optimal value for this SCM.

## A.3 ENVIRONMENT WITH A LARGE NUMBER OF VARIABLES

This example highlights a key advantage of CoCa-BO: it can yield significant gains in sample efficiency over standard CoBO, even when both methods converge to the same optimum. The advantage stems from using the causal graph to identify and remove superfluous variables from the policy scope. The resulting reduction in dimensionality often accelerates learning. We demonstrate this in a challenging, high-dimensional setting with 57 contextual and 29 intervenable variables, where the improvement from a simplified scope outweighs the cost of searching over the policy space. The underlying Structural Causal Model is defined as follows.

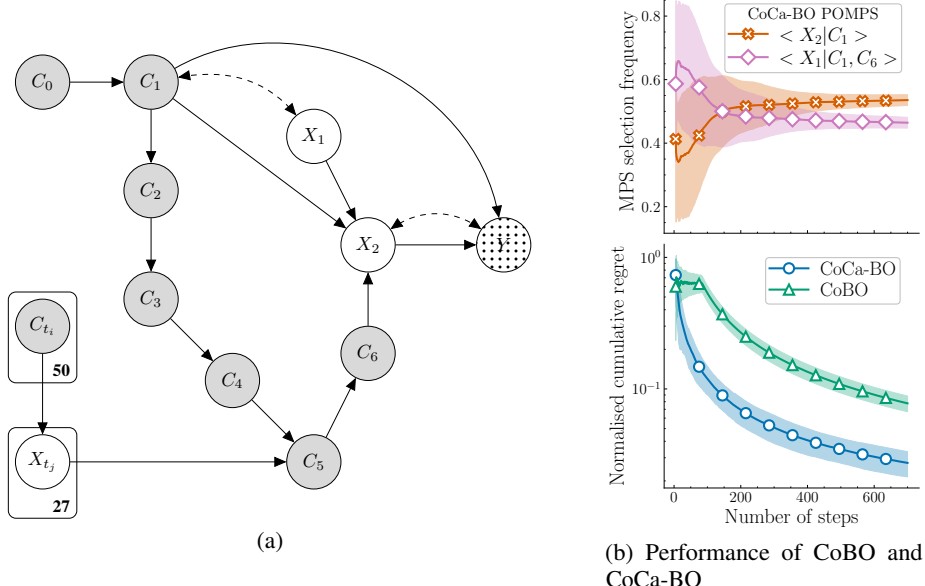

(a)

(b) Performance of CoBO and CoCa-BO

Figure 6: (a) A causal graph with a large number of variables. The rectangular nodes represent plates of variables. The number below the rectangles represents the number of variables within each plate. Variables $C_{t_i}$ and $X_{t_j}$ over-specify the environment and are redundant for optimising the expected value of $Y$. (b) Both methods converge, but CoCa-BO has lower cumulative cost due to smaller policy scopes ($\{(X_1; C_1, C_6), (X_2; C_1)\}$.

$$U_1, U_2 \sim \text{Uniform}(-1, 1), \qquad C_0 \sim \mathcal{N}(0, 0.2)$$
$$X_1 \sim \mathcal{N}(U_1, 0.1), \qquad C_1 \sim \mathcal{N}(C_0 - U_1, 0.1)$$
$$C_2 \sim \mathcal{N}(C_1, 0.1), \qquad C_3 \sim \mathcal{N}(C_2, 0.1) \qquad C_4 \sim \mathcal{N}(C_3, 0.1)$$

$$C_{t_i} \sim \mathcal{N}(0, 0.2) \qquad \forall i \in \{1, \ldots, 50\} \qquad \bar{C}_{50} = \frac{1}{50} \sum_{i=1}^{50} C_{t_i}$$

$$X_{t_j} \sim \text{Uniform}\left[-1, 1 + 0.1\bar{C}_{50}\right] \qquad \forall j \in \{1, \ldots, 27\} \qquad \bar{X}_{27} = \frac{1}{27} \sum_{j=1}^{27} X_{t_j}$$

$$C_5 \sim \mathcal{N}\left(C_4 + 0.01\bar{X}_{27}, 0.1\right) \qquad C_6 \sim \mathcal{N}(C_5, 0.1)$$
$$X_2 \sim \mathcal{N}(0.5(C_1 + C_6) + X_1 + 0.3|U_2|, 0.1)$$
$$Y \sim \mathcal{N}(\cos(C_1 - X_2) + 0.1U_2, 0.01)$$

We also provide the causal graph corresponding to the above SCM in 6a, where we use rectangular nodes to represent the plate notation (Koller & Friedman, 2009). This example has purely contextual variables $C_{t_i}$ and $X_{t_j}$ that only influence the rest of the system only through $C_5$. It is not necessary to control or observe these variables to achieve highest expectation of the target variable $Y$. This is reflected in the POMPSs $\{(X_1; C_1, C_6), (X_2; C_1)\}$ corresponding the causal graph. Both POMPSs contain noticeably fewer variables than the policy scope considered by CoBO $(X_1, X_2, X_{t_{1:27}}; C_{0:6}, C_{t_{1:50}})$ where $X_{m:n} = \{X_i \mid m \leq i \leq n\}$.

One may note that this SCM is similar to the SCM from A.2. Indeed, the structural equations for $Y$ are the same. The same derivations apply to this SCM, proving that CoBO can achieve the optimum, which is reflected in Fig. 6b, which shows that both methods converge to the optimum, yielding a decreasing trend in the normalised cumulative regret. Moreover, CoCa-BO converges faster as it utilises the causal graph and derives policies with a smaller scope. As we see, this significantly improves the convergence speed over CoBO even if it achieves the optimum.

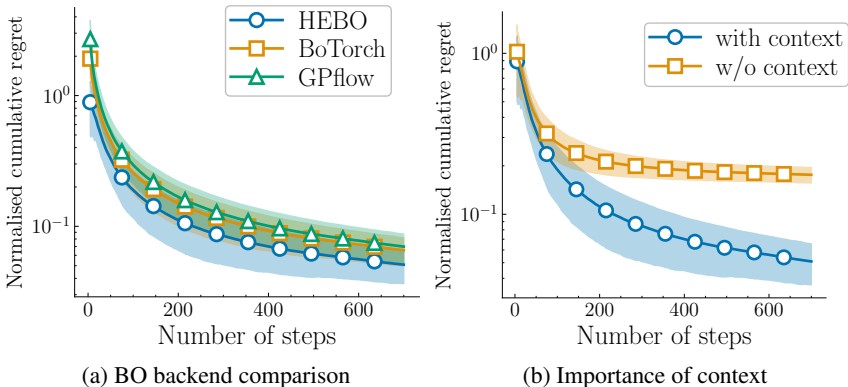

Figure 7: Left: CoCa-BO performance under different BO implementations. Right: CoCa-BO performance with contextual information preserved versus discarded.

### A.4 ROBUSTNESS TO NOISE

We note that the total variance of $Y$ under the optimal policy is actually higher than the variance of $\epsilon_Y$ due to an unobserved confounder $U_2$, which has a variance of $1/3$. Thus, $\text{var}_\pi(Y) = \frac{10^{-2}}{3} + \text{var}(\epsilon_Y)$ under the optimal policy $\pi$. We emphasise that $\text{var}_{\pi'}(Y)$ under any other policy $\pi'$ is greater than or equal to $\text{var}_\pi(Y)$.

### A.5 ROBUSTNESS TO GP BACKEND AND IMPORTANCE OF CONTEXT

This subsection examines the effect of the underlying BO optimisation on our method. We ablate the backend across HEBO (Cowen-Rivers et al. (2022)), BoTorch (Balandat et al. (2020)), and GPflow (Matthews et al. (2017)). HEBO and BoTorch utilise GPyTorch (Gardner et al. (2018)) for GP inference, whereas GPflow provides its own TensorFlow-based implementation (Abadi et al. (2015)).

We conduct the study on the SCM described in Appendix A.2 and summarise the results in Figure 7a. We observe that our method converges regardless of the underlying BO routine, though HEBO appears slightly better.

In the same environment, we ablate context by removing the contextual variables from all POMPS while preserving the interventional variables and the hierarchical design of CoCa-BO. The results in Figure 7b show that, without contextual information, the algorithm quickly converges to a suboptimal solution and cannot escape it, due to the missing context.

## B    MAIN THEORETICAL RESULTS

### B.1    DESCRIPTION OF THE SETTING AND THE ALGORITHM

Our algorithm, described in Algorithm 2, does not distinguish between two POMPSs that share the same sets of intervenable and contextual variables. Hence, we say that two MPSs $\mathcal{S}$ and $\mathcal{S}'$ are equivalent if $\mathbf{X}(\mathcal{S}) = \mathbf{X}(\mathcal{S}')$ and $\mathbf{C}(\mathcal{S}) = \mathbf{C}(\mathcal{S}')$. Let $m$ denote the number of distinct equivalence classes of POMPSs. We index these equivalence classes by $i \in \{1, \ldots, m\}$. For the $i$-th equivalence class, we denote its intervenable variables by $\mathbf{X}(i)$ and its contextual variables by $\mathbf{C}(i)$. We denote by $P_i$ the distribution of the random vector $\mathbf{C}(i)$.

We assume that the reward obtained under a POMPS $\mathcal{S}$ belonging to the $i$-th class is given by

$$Y = f_i\big(\mathbf{X}(i), \mathbf{C}(i)\big) + \varepsilon_i,$$

where $f_i : \mathfrak{X}_{\mathbf{X}(i)} \times \mathfrak{X}_{\mathbf{C}(i)} \to \mathbb{R}$ and $\varepsilon_i$ is independent Gaussian noise, $\varepsilon_i \sim \mathcal{N}(0, \sigma^2)$. Furthermore, for every equivalence class, we assume $f_i \sim \mathcal{GP}(0, k_i)$ independently across $i$, where $k_i$ are known covariance functions such that $\kappa_i := \sup_{\mathbf{x}, \mathbf{c}, \mathbf{x}', \mathbf{c}'} k_i((\mathbf{x}, \mathbf{c}), (\mathbf{x}', \mathbf{c}')) < \infty$.

This means that we assume that the following arrays of independent random objects are generated by the nature

$$
\begin{aligned}
i = 1: \quad & f_1; \quad (\mathbf{C}_{1,1}, \mathbf{C}_{1,2}, \mathbf{C}_{1,3}, \ldots, \mathbf{C}_{1,\ell}, \ldots) \quad && \sim \mathcal{GP}(0, k_1) \otimes P_1^{\otimes \mathbb{N}^*}, \\
i = 2: \quad & f_2; \quad (\mathbf{C}_{2,1}, \mathbf{C}_{2,2}, \mathbf{C}_{2,3}, \ldots, \mathbf{C}_{2,\ell}, \ldots) \quad && \sim \mathcal{GP}(0, k_2) \otimes P_2^{\otimes \mathbb{N}^*}, \\
& \vdots \quad \vdots \qquad\qquad \vdots && \qquad \vdots \\
i = m: \quad & f_m; \quad (\mathbf{C}_{m,1}, \mathbf{C}_{m,2}, \mathbf{C}_{m,3}, \ldots, \mathbf{C}_{m,\ell}, \ldots) \quad && \sim \mathcal{GP}(0, k_m) \otimes P_m^{\otimes \mathbb{N}^*}, \\
& \qquad (\varepsilon_1, \varepsilon_2, \ldots, \varepsilon_t, \ldots) && \sim \mathcal{N}(0, \sigma^2)^{\otimes \mathbb{N}^*}.
\end{aligned}
$$

Here, for a probability distribution $Q$, we denoted by $Q^{\otimes \mathbb{N}^*}$ the probability distribution of the sequence $(Z_i)_{i \in \mathbb{N}^*}$ of independent random variables drawn from the same distribution $Q$. We denote the contextual best reward for arm $i$ by

$$
f_i^*(\mathbf{c}) = \sup_{\mathbf{x} \in \mathfrak{X}_{\mathbf{X}(i)}} f_i(\mathbf{x}, \mathbf{c}),
$$

and the corresponding contextual best action by

$$
\mathbf{x}_i^*(\mathbf{c}) = \arg \max_{\mathbf{x} \in \mathfrak{X}_{\mathbf{X}(i)}} f_i(\mathbf{x}, \mathbf{c}),
$$

which we refer to as the optimal policy of arm $i$.

Throughout this section, we will use the notation $\mathbb{P}(Z \mid \boldsymbol{f})$ and $\mathbb{E}[Z \mid \boldsymbol{f}]$ to denote, respectively, the conditional probability and the conditional expectation of a random vector $Z$ given $\boldsymbol{f} = (f_1, \ldots, f_m)$. The conditioning is understood with respect to the Gaussian process $\boldsymbol{f} = (f_1, \ldots, f_m)$, which means that the integration is carried out over all realisations of the noise variables $\varepsilon_t$.

An agent, without prior knowledge of the functions $f_i$, interacts with this bandit in a sequential manner. At each time step $t$:

1. the agent chooses $i_t \in [m]$,
2. the agent observes $\mathbf{c}_t \sim P_{i_t}$,
3. the agent selects $\mathbf{x}_t \in \mathfrak{X}_{\mathbf{X}(i_t)}$,
4. the agent observes $y_t = f_{i_t}(\mathbf{x}_t, \mathbf{c}_t) + \varepsilon_t$, where $\varepsilon_t \overset{iid}{\sim} \mathcal{N}(0, \sigma^2)$.

We use $\mathbf{x}_{i,s}$ and $\mathbf{c}_{i,s}$ to refer to the interventional and contextual values of the $i^{\text{th}}$ equivalence class of POMPSs when it has been selected for the $s^{\text{th}}$ time. Additionally, we refer to an equivalence class as an *arm*, reflecting the similarity of our problem to the $m$-armed bandit setting. Each arm $i$ has $f_i$ as its expected payout function, where $\mathfrak{X}_{\mathbf{X}(i)}$ is assumed to be convex and compact.

In our causal optimisation framework, contexts for each arm are sampled i.i.d. from a fixed distribution. A key aspect of our setting is that the agent actively selects which contextual variables to observe, a decision determined by the chosen POMPS. The agent's objective is to maximize (in expectation) the cumulative reward $\sum_{t=1}^{T} y_t$ over an unknown time horizon $T$.

Our proposed algorithm relies on the upper confidence bound (UCB) as the selection criterion for the evaluation point. Recall that the posterior mean and variance of the GP are denoted by $\mu_{i,s}(\mathbf{x}, \mathbf{c})$ and $\sigma_{i,s}(\mathbf{x}, \mathbf{c})$, respectively, after the $i^{\text{th}}$ arm has been played $s$ times. The UCB at the point $(\mathbf{x}, \mathbf{c})$ after $s$ plays is defined as

$$
\text{UCB}_{i,\ell}(\mathbf{x}, \mathbf{c}) = \mu_{i,\ell-1}(\mathbf{x}, \mathbf{c}) + \sqrt{\beta_i(\ell)} \, \sigma_{i,\ell-1}(\mathbf{x}, \mathbf{c}), \tag{5}
$$

for some tuning parameters $\beta_i(\ell)$. We summarise our method in Algorithm 2, where $\beta_{i,s}$ and $\pi_s$ are free parameters to be specified later.

**Regret Definition** Let $\mu_i = \mathbb{E}[f_i^*(\mathbf{C}_{i,1}) \mid f_i] = \int \max_{\mathbf{x}} f_i(\mathbf{x}, \mathbf{c}) P_i(d\mathbf{c})$ and $I^* = \text{argmax}_{i \in [m]} \mu_i$. Clearly, $\mu_i$ depends on $f_i$, whereas $R_T$ and $I^*$ depend on $\boldsymbol{f}$, but for the sake of simplicity of writing, this dependence will not be highlighted in the notation. We also write $\mu^* = \max_{i \in [m]} \mu_i$. The regret then is defined as:

$$
R_T = \sum_{t=1}^{T} \left\{ \mu^* - \mathbb{E}[f_{i_t}(\mathbf{X}_t, \mathbf{C}_t) \mid \boldsymbol{f}, \mathcal{D}^{t-1}] \right\}. \tag{6}
$$

---

**Algorithm 2** Intervention selection algorithm UCB-BO

**Input:** $m :=$ number of equivalence classes of MPSs from $\mathbb{S}^*$
**Parameters:** $(\beta_i(n))_{i\in[m];n\in\mathbb{N}^*}$, $(\rho_i(n))_{i\in[m];n\in\mathbb{N}^*}$.
**for** $t = 1$ to ... **do**
   **if** $t \leq m$ **then**
      Select an equivalence class: $i_t := t$.
      Observe context $\mathbf{c}_t = \mathbf{c}_{i_t,1}$.
      Select an action: $\mathbf{x}_t = \mathbf{x}_{i_t,1} = \arg\max_{\mathbf{x}\in\mathfrak{X}_{\mathbf{X}(i_t)}} \text{UCB}_{i_t,1}(\mathbf{x},\mathbf{c}_{i_t,1})$. (See Eq. (5) for UCB.)
      Set: $\bar{U}_{i_t}(1) := \text{UCB}_{i_t,1}(\mathbf{x}_{i_t,1},\mathbf{c}_{i_t,1})$; $n_{i_t} = 1$.
      Observe: $y_t$
      Update the posterior for the $i_t$ Gaussian process based on $\mathbf{x}_{i_t,1}, \mathbf{c}_{i_t,1}$, and $y_t$.
   **else**
      Select an equivalence class: $i_t = \arg\max_{i\in[m]} \left(\bar{U}_i(n_i) + \frac{\rho_i(n_i)}{\sqrt{n_i}}\right)$.
      Update: $n_{i_t} = n_{i_t} + 1$.
      Observe context: $\mathbf{c}_t = \mathbf{c}_{i_t,n_{i_t}}$
      Select an action: $\mathbf{x}_t = \mathbf{x}_{i_t,n_{i_t}} = \arg\max_{\mathbf{x}\in\mathfrak{X}_{\mathbf{X}_{i_t}}} \text{UCB}_{i_t,n_{i_t}}(\mathbf{x},\mathbf{c}_t)$
      Observe: $y_t$
      Update the posterior for the $i_t$ Gaussian process based on $\mathbf{x}_t, \mathbf{c}_t$, and $y_t$.
      Set: $\bar{U}_{i_t}(n_{i_t}) = \frac{(n_{i_t}-1)}{n_{i_t}}\bar{U}_{i_t}(n_{i_t}-1) + \frac{1}{n_{i_t}}\text{UCB}_{i_t,n_{i_t}}(\mathbf{x}_t,\mathbf{c}_t)$
   **end if**
   **end for**

---

The regret $R_T$ can be decomposed into two components:

$$R_T = \sum_{t=1}^{T}\left\{ \underbrace{\mu^* - \mu_{i_t}}_{\text{Arm Selection Regret}} + \underbrace{\mu_{i_t} - \mathbb{E}[f_{i_t}(\mathbf{X}_t,\mathbf{C}_t) \mid \boldsymbol{f},\mathcal{D}^{t-1}]}_{\text{Context-averaged Action Selection Regret}} \right\},$$

The context-averaged action selection regret measures how much the mean reward $f_{i_t}$ of the action $\mathbf{x}_t$ chosen by the agent deviates from the mean reward of the best possible action $\mathbf{x}_i^*(\mathbf{c}_{i_t})$, averaged over all values of the context. We note that it is localised to the chosen arm. The arm selection regret compares the rewards of each arm if the agent plays according to the optimal policy of that arm.

## B.2 Proof of the main result

Our next lemmas motivate the choice of the index for arms in Algorithm 2. Recall that $\mathbf{X}_{i,\ell}$ is an element of $\mathfrak{X}_{\mathbf{X}(i)}$ that maximises $\mathbf{x} \mapsto \text{UCB}_{i,\ell}(\mathbf{x}, \mathbf{C}_{i,\ell})$.

**Lemma 1.** *Let $\delta \in (0,1)$ and $(\pi_n)_{n\in\mathbb{N}^*}$ be a nondecreasing sequence of positive numbers such that $\sum_{n\in\mathbb{N}^*} \pi_n^{-1} = 1$. Let $\mathcal{N}_{i,n}$ be an $\varepsilon$-net of $\mathfrak{X}_{\mathbf{X}(i)}$ in $\ell_1$-norm, with $\varepsilon = 1/(m\pi_n r_i d_i)$. If for all $n \in \mathbb{N}^*$,*

$$\beta_i(n) \geq 2\log(2/(m\delta)) + 4d_i\log_+(mr_id_i\pi_n),$$

*there is an event $\mathcal{A}_i(\delta)$ of probability at least $1 - \delta$ such that on $\mathcal{A}_i(\delta)$, the following properties hold*

$$f_i(\mathbf{X}_{i,\ell},\mathbf{C}_{i,\ell}) \geq \text{UCB}_{i,\ell}(\mathbf{X}_{i,\ell},\mathbf{C}_{i,\ell}) - 2\sqrt{\beta_i(\ell)}\,\sigma_{i,\ell-1}(\mathbf{X}_{i,\ell},\mathbf{C}_{i,\ell}); \text{ for all } \ell \in \mathbb{N}^*, \quad (7)$$

$$f_i^*(\mathbf{C}_{i,\ell}) \leq \text{UCB}_{i,\ell}(\mathbf{X}_{i,\ell},\mathbf{C}_{i,\ell}) + \frac{\varphi}{m\pi_\ell}\sqrt{\log(2\bar{d}_i\psi/\delta)}; \quad \text{for all } \mathbf{x} \in \mathfrak{X}_{\mathbf{X}(i)}; \ \ell \in \mathbb{N}^*. \quad (8)$$

*Proof.* We define the events

$$\Omega_0 = \left\{|f_i(\mathbf{x},\mathbf{c}) - f_i(\mathbf{x}',\mathbf{c})| \leq \varphi\sqrt{\log(2\bar{d}_i\psi/\delta)}\,\|\mathbf{x}-\mathbf{x}'\|_1, \text{ for all } \mathbf{x},\mathbf{x}' \in \mathfrak{X}_{\mathbf{X}(i)},\ \mathbf{c} \in \mathfrak{X}_{\mathbf{C}(i)}\right\},$$

$$\Omega_{\ell,1} = \left\{f_i(\mathbf{x},\mathbf{C}_{i,\ell}) \leq \text{UCB}_{i,\ell}(\mathbf{x},\mathbf{C}_{i,\ell}); \quad \text{for all } \mathbf{x} \in \mathcal{N}_{i,\ell}\right\},$$

$$\Omega_{\ell,2} = \left\{f_i^*(\mathbf{C}_{i,\ell}) \geq \text{UCB}_{i,\ell}(\mathbf{X}_{i,\ell},\mathbf{C}_{i,\ell}) - 2\sqrt{\beta_i(\ell)}\,\sigma_{i,\ell-1}(\mathbf{X}_{i,\ell},\mathbf{C}_{i,\ell})\right\}.$$

Assumption 2 implies that $\mathbb{P}(\Omega_0) \geq 1 - \delta/2$. Furthermore, we recall that for any $(\mathbf{x}, \mathbf{c}) \in \mathfrak{X}_{\mathbf{X}(i)} \times \mathfrak{X}_{\mathbf{C}(i)}$ the conditional distribution of $f_i(\mathbf{x}, \mathbf{c})$ given $\mathcal{D}_{i,\ell-1}$ is Gaussian with mean $\mu_{i,\ell-1}(\mathbf{x}, \mathbf{c})$ and variance $\sigma^2_{i,\ell-1}(\mathbf{x}, \mathbf{c})$. This implies that

$$\mathbb{P}\Big(f_i(\mathbf{x}, \mathbf{c}) \leq \mu_{i,\ell-1}(\mathbf{x}, \mathbf{c}) + \sqrt{2\log(1/\delta_\ell)}\,\sigma_{\ell-1}(\mathbf{x}, \mathbf{c}) \,\Big|\, \mathcal{D}_{i,\ell-1}\Big) \geq 1 - \delta_\ell/2,$$

for every $\delta_\ell \in (0,1)$. Note that the above inequality holds not only for any deterministic $(\mathbf{x}, \mathbf{c})$, but also for every random pair $(\mathbf{x}, \mathbf{c})$ provided that it is independent of $f_i$ given $\mathcal{D}_{i,\ell-1}$. Therefore, applying this inequality to all pairs $\{(\mathbf{x}, \mathbf{C}_{i,\ell}) : \mathbf{x} \in \mathcal{N}_{i,\ell}\}$ and taking the union bound, we get

$$\mathbb{P}\Big(f_i(\mathbf{x}, \mathbf{C}_{i,\ell}) \leq \mu_{i,\ell-1}(\mathbf{x}, \mathbf{C}_{i,\ell}) + \sqrt{\beta_i(\ell)}\,\sigma_{i,\ell-1}(\mathbf{x}, \mathbf{C}_{i,\ell}); \;\forall \mathbf{x} \in \mathcal{N}_{i,\ell} \,\Big|\, \mathcal{D}_{i,\ell-1}\Big) \geq 1 - \frac{|\mathcal{N}_{i,\ell}|\delta_\ell}{2},$$

as soon as $\beta_i(\ell) \geq 2\log(1/\delta_\ell)$. Taking into account the definition of $\mathrm{UCB}_{i,\ell}$, we arrive at $\mathbb{P}(\Omega_{\ell,1}|\mathcal{D}_{i,\ell-1}) \geq 1 - |\mathcal{N}_{i,\ell}|\delta_\ell/2$. Since this inequality holds for any set of values taken by the data $\mathcal{D}_{i,\ell-1}$, we get

$$\mathbb{P}(\Omega_{\ell,1}) \geq 1 - (\delta/4\pi_\ell), \quad \text{provided that} \quad \beta_i(\ell) \geq 2\log(2\pi_\ell|\mathcal{N}_{i,\ell}|/\delta).$$

Since $\mathbf{X}_{i,\ell}$ is the maximiser in $\mathfrak{X}_{\mathbf{X}(i)}$ of $\mathbf{x} \mapsto \mathrm{UCB}_{i,\ell}(\mathbf{x}, \mathbf{C}_{i,\ell})$, it is clearly independent of $f_i$ conditionally to $\mathcal{D}_{i,\ell-1}$. Therefore, the argument used above can be repeated to infer that the following inequalities hold with probability $\geq 1 - \delta/(4\pi_\ell)$:

$$f_i(\mathbf{X}_{i,\ell}, \mathbf{C}_{i,\ell}) \geq \mu_{i,\ell-1}(\mathbf{X}_{i,\ell}, \mathbf{C}_{i,\ell}) - \sqrt{2\log(2\pi_\ell/\delta)}\,\sigma_{i,\ell-1}(\mathbf{X}_{i,\ell}, \mathbf{C}_{i,\ell})$$
$$\geq \mathrm{UCB}(\mathbf{X}_{i,\ell}, \mathbf{C}_{i,\ell}) - 2\sqrt{\beta_i(\ell)}\,\sigma_{i,\ell-1}(\mathbf{X}_{i,\ell}, \mathbf{C}_{i,\ell}),$$

provided that $\beta_i(\ell) \geq 2\log(2\pi_\ell/\delta)$. We conclude that as soon as $\beta_i(\ell) \geq 2\log(2\pi_\ell|\mathcal{N}_{i,\ell}|/\delta)$, we have $\mathbb{P}(\mathcal{A}_i(\delta)) \geq 1 - \delta$ for

$$\mathcal{A}_i(\delta) = \Omega_0 \cap \Big\{ \bigcap_{\ell=1}^{\infty} \big(\Omega_{\ell,1} \cap \Omega_{\ell,2}\big)\Big\}.$$

The fact that (7) holds on $\mathcal{A}(\delta)$ is obvious. To show that (8) holds as well, it suffices to notice that for all $\mathbf{x} \in \mathfrak{X}_{\mathbf{X}(i)}$ and for all $\ell \in \mathbb{N}^*$,

$$f_i(\mathbf{x}, \mathbf{C}_{i,\ell}) \leq \max_{\mathbf{x} \in \mathcal{N}_{i,\ell}} f_i(\mathbf{x}, \mathbf{C}_{i,\ell}) + \varphi d_i r_i \sqrt{\log(2\bar{d}_i\psi/\delta)}\,(m\pi_\ell r_i d_i)^{-1}$$
$$\leq \mathrm{UCB}_{i,\ell}(\mathbf{x}, \mathbf{C}_{i,\ell}) + \varphi\sqrt{\log(2d_i\psi/\delta)}\,(m\pi_\ell)^{-1}$$
$$\leq \mathrm{UCB}_{i,\ell}(\mathbf{X}_{i,\ell}, \mathbf{C}_{i,\ell}) + \varphi\sqrt{\log(2\bar{d}_i\psi/\delta)}\,(m\pi_\ell)^{-1}.$$

The first inequality above follows from the fact that $\Omega_0$ is realised, the second inequality follows from $\Omega_{\ell,1}$ and the third inequality is due to the fact that $\mathbf{X}_{i,\ell}$ is the maximiser of $\mathbf{x} \mapsto \mathrm{UCB}_{i,\ell}(\mathbf{x}, \mathbf{C}_{i,\ell})$. Since we assumed that $\mathfrak{X}_{\mathbf{X}(i)} \subset [0, r_i]^{d_i}$, we have $|\mathcal{N}_{i,\ell}| \leq (r_i d_i/\varepsilon)^{d_i} \vee 1 = (m r_i^2 d_i^2 \pi_\ell)^{d_i} \vee 1$. This implies that it suffices to choose $\beta_i(\ell)$ satisfying

$$\beta_i(\ell) \geq 2\log(2/(m\delta)) + 4d_i\log_+(m r_i d_i \pi_\ell).$$

This completes the proof of the lemma. $\qquad\square$

**Lemma 2.** *Let $(\pi_n)_{n \in \mathbb{N}^*}$ be a nondecreasing sequence of positive numbers such that $\sum_{n \in \mathbb{N}^*} \pi_n^{-1} = 1$. Define*

$$\bar{U}_i(n) = \frac{1}{n}\sum_{\ell=1}^n \mathrm{UCB}_{i,\ell}(\mathbf{X}_{i,\ell}, \mathbf{C}_{i,\ell}), \qquad \bar{\mu}_i(n) = \frac{1}{n}\sum_{\ell=1}^n f_i^*(\mathbf{C}_{i,\ell}).$$

*If (7) and (8) are satisfied for some $\delta \in (0,1)$, then for every $n \in \mathbb{N}^*$,*

$$\bar{\mu}_i(n) - \frac{\varphi\sqrt{\log(2d_i\psi/\delta)}}{mn} \leq \bar{U}_i(n) \leq \bar{\mu}_i(n) + 2\sqrt{\frac{2\kappa_i\beta_i(n)\gamma_i(n)}{n\log(1+\sigma^{-2}\kappa_i)}}.$$

*Proof.* From (8), we infer that

$$f_i^*(\mathbf{C}_{i,\ell}) \leq \mathrm{UCB}_{i,\ell}(\mathbf{X}_{i,\ell}, \mathbf{C}_{i,\ell}) + \frac{\varphi}{m\pi_\ell}\sqrt{\log(2\bar{d}_i\psi/\delta)}.$$

Averaging over $\ell = 1, \ldots, n$, and using the inequality $\sum_{\ell=1}^n 1/\pi_\ell \leq 1$, we obtain

$$\bar{\mu}_i(n) \leq \bar{U}_i(n) + \frac{\varphi d_i r_i}{mn}\sqrt{\log(2\bar{d}_i\psi/\delta)}.$$

To get a lower bound on $\bar{\mu}_i(n)$, we average over $\ell = 1, \ldots, n$ inequalities (7), then apply the Cauchy-Schwarz inequality, to obtain

$$\begin{aligned}
\bar{\mu}_i(n) &\geq \bar{U}_i(n) - \frac{2}{n}\sum_{\ell=1}^n \sqrt{\beta_i(\ell)}\,\sigma_{i,\ell-1}(\mathbf{X}_{i,\ell}, \mathbf{C}_{i,\ell}) \\
&\geq \bar{U}_i(n) - \frac{2\sqrt{\beta_i(n)}}{n}\sum_{\ell=1}^n \sigma_{i,\ell-1}(\mathbf{X}_{i,\ell}, \mathbf{C}_{i,\ell}) \\
&\geq \bar{U}_i(n) - \frac{2\sqrt{\beta_i(n)}}{\sqrt{n}}\left\{\sum_{\ell=1}^n \sigma_{i,\ell-1}^2(\mathbf{X}_{i,\ell}, \mathbf{C}_{i,\ell})\right\}^{1/2}.
\end{aligned} \tag{9}$$

The definition of $\sigma_{i,\ell}^2(\mathbf{x}, \mathbf{c})$ as the posterior variance of $f_i(\mathbf{x}, \mathbf{c})$ given $\mathcal{D}_{i,\ell-1}$ and formula (1) imply that $\sigma_{i,\ell}^2(\mathbf{x}, \mathbf{c}) \leq \kappa_i$ for every $(\mathbf{x}, \mathbf{c})$. Therefore, the increasingness of the function $x \mapsto x/\log(1+x)$ yields

$$\frac{\sigma^{-2}\sigma_{i,\ell-1}^2(\mathbf{x}, \mathbf{C}_{i,\ell})}{\log\left(1 + \sigma^{-2}\sigma_{i,\ell-1}^2(\mathbf{x}, \mathbf{C}_{i,\ell})\right)} \leq \frac{\sigma^{-2}\kappa_i}{\log(1 + \sigma^{-2}\kappa_i)}.$$

This can be rewritten as

$$\sigma_{i,\ell-1}^2(\mathbf{X}_{i,\ell}, \mathbf{C}_{i,\ell}) \leq \frac{\kappa_i \log\left(1 + \sigma^{-2}\sigma_{i,\ell-1}^2(\mathbf{X}_{i,\ell}, \mathbf{C}_{i,\ell})\right)}{\log(1 + \sigma^{-2}\kappa_i)}, \qquad \forall \ell \in \mathbb{N}^*.$$

Therefore, we have

$$\begin{aligned}
\sum_{\ell=1}^n \sigma_{i,\ell-1}^2(\mathbf{X}_{i,\ell}, \mathbf{C}_{i,\ell}) &\leq \frac{\kappa_i}{\log(1 + \sigma^{-2}\kappa_i)}\sum_{\ell=1}^n \log\left(1 + \sigma^{-2}\sigma_{i,\ell-1}^2(\mathbf{X}_{i,\ell}, \mathbf{C}_{i,\ell})\right) \\
&\leq \frac{2\kappa_i\gamma_i(n)}{\log(1 + \sigma^{-2}\kappa_i)}.
\end{aligned} \tag{10}$$

The last inequality above follows from the definition of $\gamma_n$ and (Srinivas et al., 2010, Lemma 5.3).

Combining (9) and (10), we arrive at

$$\bar{U}_i(n) \leq \bar{\mu}_i(n) + 2\sqrt{\frac{2\kappa_i\beta_i(n)\gamma_i(n)}{n\log(1 + \sigma^{-2}\kappa_i)}}.$$

This completes the proof of the lemma. $\qquad\qquad\square$

**Lemma 3.** *Let $\delta \in (0, 1)$ and let $\mathfrak{X}_{\mathbf{C}(i)}$ be convex and compact, included in $[0, r_i']^{d_i'}$, for some $d_i' \in \mathbb{N}^*$ and some $r_i' \geq 0$. Let us denote*

$$\lambda_i'(\delta) = d_i' r_i' \varphi \log(2(d_i'\psi + 2\pi_n)/\delta). \tag{11}$$

*There exists an event $\mathcal{A}_i'(\delta)$ of probability at least $1 - \delta$ such that on $\mathcal{A}_i'(\delta)$, it holds*

$$\left|\frac{1}{n}\sum_{\ell=1}^n f_i^*(\mathbf{C}_{i,\ell}) - \mu_i\right| \leq \frac{\lambda_i'(\delta)}{\sqrt{2n}}, \qquad \forall n \in \mathbb{N}^*.$$

*Proof.* If we set $\Delta f_i^* = \max_{\mathbf{c}} f_i^*(\mathbf{c}) - \min_{\mathbf{c}} f_i^*(\mathbf{c})$, the Höffding inequality implies that

$$\mathbb{P}\left(\left|\frac{1}{n}\sum_{\ell=1}^{n} f_i^*(\mathbf{C}_{i,\ell}) - \mu_i\right| \leq \Delta f_i^*\sqrt{\frac{\log(2/\delta_1)}{2n}}\,\bigg|\,\boldsymbol{f}\right) \geq 1 - \delta_1, \quad \forall i \in [m].$$

Choosing $\delta_1 = \delta/(2\pi_n)$ and using the Bonferroni inequality, we get

$$\mathbb{P}\left(\left|\frac{1}{n}\sum_{\ell=1}^{n} f_i^*(\mathbf{C}_{i,\ell}) - \mu_i\right| \leq \Delta f_i^*\sqrt{\frac{\log(4\pi_n/\delta)}{2n}} \text{ for all } n \in \mathbb{N}^*\,\bigg|\,\boldsymbol{f}\right) \geq 1 - \delta/2. \tag{12}$$

On the other hand, since $\mathfrak{X}_{\mathbf{C}(i)}$ is convex and compact, included in $[0, r_i']^{d_i'}$, Assumption 2 implies that there exist kernel dependent positive constants $\varphi, \psi$ such that, with probability at least $1 - \delta/2$,

$$|f_i(\mathbf{x}, \mathbf{c}) - f_i(\mathbf{x}, \mathbf{c}')| \leq \varphi\sqrt{\log(2d_i'\psi/\delta)}\,\|\mathbf{c} - \mathbf{c}'\|_1; \quad \text{for all } \mathbf{x} \in \mathfrak{X}_{\mathbf{X}(i)}, \mathbf{c}, \mathbf{c}' \in \mathfrak{X}_{\mathbb{C}(i)}.$$

Hence, with probability at least $1 - \delta/2$, we have

$$\Delta f_i^* \leq d_i' r_i'.\varphi\sqrt{\log(2d_i'\psi/\delta)}.$$

Combining with (12), this implies that on an event of probability at least $1 - \delta$, denoted by $\mathcal{A}_i'(\delta)$, we have, for any $n \in \mathbb{N}^*$,

$$\left|\frac{1}{n}\sum_{\ell=1}^{n} f_i^*(\mathbf{C}_{i,\ell}) - \mu_i\right| \leq d_i' r_i' \varphi\sqrt{\log(2d_i'\psi/\delta)}\sqrt{\frac{\log(4\pi_n/\delta)}{2n}}$$

$$\leq d_i' r_i' \varphi\frac{\log(2(d_i'\psi + 2\pi_n)/\delta)}{\sqrt{2n}} = \frac{\lambda_i'(\delta)}{\sqrt{2n}}.$$

This completes the proof of the lemma. $\qquad\square$

Our next step is to upper bound the number of times a suboptimal arm $i \notin I^*$ is played.

**Lemma 4.** *Let* $\delta \in (0, 1/2)$ *and* $(\pi_n)$ *be a positive nondecreasing sequence satisfying* $\sum_{n\in\mathbb{N}^*} 1/\pi_n = 1$. *Let* $\mathsf{C}_i = 8\kappa_i/\log(1 + \sigma^{-2}\kappa_i)$. *Assume that the parameters* $\beta_i(n)$ *and* $\rho_i(n)$ *are selected so that, for every* $i \in [m]$ *and every* $n \in \mathbb{N}^*$,

$$\beta_i(n) \geq 2\log(4/\delta) + 4d_i\log(mr_id_i\pi_n),$$
$$\rho_i(n) \geq \varphi\sqrt{\log(4m\bar{d}_i\psi/\delta)} + d_i'r_i'\varphi\log\left(4m(d_i'\psi + 2\pi_n)/\delta\right).$$

*Let* $\mathcal{A}(\delta) = \bigcap_{i=1}^{m}\left(\mathcal{A}_i(\delta/2m) \cap \mathcal{A}_i'(\delta/2m)\right)$ *be the intersection of the events defined in Lemma 1 and Lemma 3. Then* $\mathbb{P}(\mathcal{A}(\delta)) \geq 1 - \delta$ *and on* $\mathcal{A}(\delta)$, *the number of times* $n_i = n_i(T)$ *a suboptimal arm with a gap* $\Delta_i > 0$ *has been played over the horizon* $T$ *satisfies*

$$\sqrt{n_i(T)} \leq \frac{\sqrt{\mathsf{C}_i\beta_i(n_i)\gamma_i(n_i)} + 2\rho_i(n_i)}{\Delta_i}\bigvee 1, \quad \forall T \geq 1, \forall i \notin I^*.$$

*Proof.* The fact that $\mathbb{P}(\mathcal{A}(\delta)) \geq 1 - \delta$ follows from Lemma 1, Lemma 3 and the union bound. Let us use the notations

$$b_i(n) = \sqrt{\mathsf{C}_i\beta_i(n)\gamma_i(n)}$$

and $\lambda_i'$ as defined by (11). For all pairs $(T, i)$ such that $n_i(T) = 1$, the desired inequality is obviously satisfied. Consider now the pairs $(T, i)$ such that $n_i(T) \geq 2$. Let $t_i$ be the last time the arm $i$ has been played among the first $T$ rounds. To ease notation, let us set $n_i = n_i(t_i) = n_i(T)$. Let $j = j(\boldsymbol{f})$ be an optimal arm, that is, an integer from $[m]$ maximising $i \mapsto \mu_i$. Since at round $t_i$ arm $i$ has been preferred to the optimal arm $j$, we have

$$\bar{U}_i(n_i) + \frac{\rho_i(n_i)}{\sqrt{n_i}} \geq \bar{U}_j(n_j) + \frac{\rho_j(n_j)}{\sqrt{n_j}}. \tag{13}$$

Under our choice of $\beta_i(n)$ and $\pi_n$, Lemma 1 and Lemma 2 applied with $\delta/(2m)$ instead of $\delta$, imply that on $\mathcal{A}(\delta)$, for all $T \geq 1$ and for all $i \notin I^*$, it holds that

$$\bar{U}_i(n_i) \leq \bar{\mu}_i(n_i) + \frac{b_i(n_i)}{\sqrt{n_i}}, \tag{14}$$

$$\bar{U}_j(n_j) \geq \bar{\mu}_j(n_j) - \frac{\varphi_j \sqrt{\log(4md_j\psi_j/\delta)}}{mn_j}. \tag{15}$$

Combining (13), (14) and (15), we check that with probability at least $1 - \delta/2$, for all $T \geq 1$ and for all $i \notin I^*$, it holds that

$$\bar{\mu}_i(n_i) + \frac{b_i(n_i) + \rho_i(n_i)}{\sqrt{n_i}} \geq \bar{\mu}_j(n_j) + \frac{\rho_j(n_j)}{\sqrt{n_j}} - \frac{\varphi_j \sqrt{\log(4md_j\psi_j/\delta)}}{mn_j}. \tag{16}$$

Since we work under $\mathcal{A}(\delta)$ that is included in $\bigcup_{i=1}^m \mathcal{A}'_i(\delta/2m)$, we have

$$|\bar{\mu}_i(n) - \mu_i| \leq \frac{\lambda'_i(\delta/2m)}{\sqrt{2n}}, \qquad \text{for all } i \in [m].$$

In conjunction with (16), this implies that for all $T \geq 1$ and for all $i \notin I^*$, $j \in I^*$,

$$\mu_i + \frac{b_i(n_i) + \rho_i(n_i)}{\sqrt{n_i}} + \frac{\lambda'_i(\delta/2m)}{\sqrt{2n_i}} \geq \mu_j + \frac{\rho_j(n_j)}{\sqrt{n_j}} - \frac{\varphi_j \sqrt{\log(4md_j\psi_j/\delta)}}{mn_j} - \frac{\lambda'_j(\delta/2m)}{\sqrt{2n_j}}.$$

In the rest of this proof, we work under this event.

When $\rho_j(n)$ is chosen so that for all $j \in [m]$,

$$\rho_j(n) \geq \frac{\varphi_j}{m} \sqrt{\frac{\log(4md_j\psi_j/\delta)}{n}} + \frac{\lambda'_j(\delta/2m)}{\sqrt{2}},$$

we infer that

$$\frac{b_i(n_i) + \rho_i(n_i) + \lambda'_i(\delta/2m)}{\sqrt{n_i}} \geq \Delta_i, \qquad \text{for all } i \notin I^*.$$

This implies that

$$\sqrt{n_i} \leq \frac{b_i(n_i) + \rho_i(n_i) + \lambda'_i(\delta/2m)}{\Delta_i} \leq \frac{b_i(n_i) + 2\rho_i(n_i)}{\Delta_i}.$$

This completes the proof of the lemma. $\qquad\square$

From now on, we use the notation

$$\mathsf{C} = \max_i \mathsf{C}_i, \quad \kappa = \max_i \kappa_i, \quad d = \max_i d_i, \quad d' = \max_i d'_i,$$

$$\beta(T) = \max_i \max_{n \leq T} \beta_i(n), \quad \rho(T) = \max_i \max_{n \leq T} \rho_i(n),$$

and

$$\gamma(T) = \max_i \gamma_i(T), \quad \bar{\gamma}(T) = \max_{(n_i):\sum_i n_i = T} \frac{1}{m} \sum_{i=1}^m \gamma_i(n_i),$$

the last maximum being over all the sequences of positive integers summing to $T$. We omit the dependence of $n_i(T)$ on the horizon $T$ and write $n_i$ for brevity. We denote $\beta_I = \max_{i \in I} \beta_{i,T}$ and upper bound $R_T$ under Lemma 4.

**Proposition 1.** *Let $\delta \in (0, 1/2)$ and $(\pi_n)$ be a positive nondecreasing sequence satisfying $\sum_{n \in \mathbb{N}^*} 1/\pi_n = 1$. Let $\mathsf{C} = 8\kappa/\log(1 + \sigma^{-2}\kappa)$. Let $\mathbf{\Delta} = (\Delta_1, \ldots, \Delta_m)$ be the vector of sub-optimality gaps defined by $\Delta_i = \mu^* - \mu_i$ and $I^* = \{i : \Delta_i = 0\}$. Let Assumption 1 and Assumption 2 be fulfilled, and assume that the parameters $\beta_i$ and $\rho_i$ of the algorithm satisfy*

$$\beta_i(n) \geq 2\log(4/\delta) + 4d_i \log(mr_id_i\pi_n),$$

$$\rho_i(n) \geq \varphi\sqrt{\log(4m\bar{d}_i\psi/\delta)} + d'_ir'_i\varphi \log\left(4m(d'_i\psi + 2\pi_n)/\delta\right)$$

*for every $i \in [m]$ and every $n \in \mathbb{N}^*$. If the UCB-BO Algorithm 2 is applied up to horizon $T$ then, with probability at least $1 - 2\delta$, the regret defined by (6) satisfies the following two inequalities*

$$R_T \leq 5\sqrt{mT}\left(\sqrt{\mathsf{C}\beta(T)\bar{\gamma}(T)} + \rho(T)\right) + \|\boldsymbol{\Delta}\|_1,$$

$$R_T \leq 2\sqrt{|I^*|T}\left(\sqrt{\mathsf{C}\beta(T)\bar{\gamma}(T)} + \rho(T)\right) + 22\left(\mathsf{C}\beta(T)\gamma(T) + \rho^2(T)\right)\sum_{i \notin I^*}\frac{1}{\Delta_i}$$

$$+ \|\boldsymbol{\Delta}\|_1 + 2m\left(\sqrt{\kappa\beta(1)} + \rho(T)\right).$$

*for all $T \in \mathbb{N}^*$.*

*Proof.* Throughout the proof, we assume that the event $\mathcal{A}(\delta)$, of probability at least $1 - \delta$, defined in Lemma 4, is realised. We will also repeatedly use the following inequalities, that are direct consequences of the Cauchy-Schwarz inequality:

$$\sum_{i \in J}\sqrt{n_i(T)} \leq \sqrt{|J|T}, \qquad \sum_{i \in J}\sqrt{n_i(T)\rho_i(n_i(T))} \leq \sqrt{mT\bar{\rho}(T)}, \quad \text{for all } J \subset [m].$$

We will also use the notation

$$c(n) = \sqrt{\mathsf{C}\beta(n)\gamma(n)} + \rho(n),$$

$$\bar{c}(n) = \sqrt{\mathsf{C}\beta(n)\bar{\gamma}(n)} + \rho(n),$$

for any $n \in \mathbb{N}^*$. Let us define the functions $f_i^{\circ}(\mathbf{x}, \mathbf{c}) = f_i^*(\mathbf{c}) - f_i(\mathbf{x}, \mathbf{c})$. With this notation, the regret $R_T$ can be written as:

$$R_T = \sum_{t=1}^{T}\left\{\mu^* - \mathbb{E}[f_{i_t}(\mathbf{X}_t, \mathbf{C}_t) \mid \boldsymbol{f}, \mathcal{D}^{t-1}]\right\}$$

$$= \sum_{t=1}^{T}\left\{\mu^* - \mu_{i_t} + \mathbb{E}[f_{i_t}^{\circ}(\mathbf{X}_t, \mathbf{C}_t) \mid \boldsymbol{f}, \mathcal{D}^{t-1}]\right\}$$

$$= \sum_{t=1}^{T}\{\mu^* - \mu_{i_t}\} + \sum_{t=1}^{T}f_{i_t}^{\circ}(\mathbf{X}_t, \mathbf{C}_t) + \sum_{t=1}^{T}\left\{\mathbb{E}[f_{i_t}^{\circ}(\mathbf{X}_t, \mathbf{C}_t) \mid \boldsymbol{f}, \mathcal{D}^{t-1}] - f_{i_t}^{\circ}(\mathbf{X}_t, \mathbf{C}_t)\right\}.$$

This leads to a decomposition of the regret into three components

$$R_T = R_{T,1} + R_{T,2} + R_{T,3}$$

with

$$R_{T,1} = \sum_{t=1}^{T}\{\mu^* - \mu_{i_t}\}, \qquad R_{T,2} = \sum_{t=1}^{T}f_{i_t}^{\circ}(\mathbf{X}_t, \mathbf{C}_t)$$

$$R_{T,3} = \sum_{t=1}^{T}\left\{\mathbb{E}[f_{i_t}^{\circ}(\mathbf{X}_t, \mathbf{C}_t) \mid \boldsymbol{f}, \mathcal{D}^{t-1}] - f_{i_t}^{\circ}(\mathbf{X}_t, \mathbf{C}_t)\right\}.$$

One can check that for any function $h$ defined on the appropriate set, we have

$$\sum_{t=1}^{T}h(i_t, \mathbf{X}_t, \mathbf{C}_t) = \sum_{i=1}^{m}\sum_{t:i_t=i}h(i_t, \mathbf{X}_t, \mathbf{C}_t) = \sum_{i=1}^{m}\sum_{\ell=1}^{n_i(T)}h(i, \mathbf{X}_{i,\ell}, \mathbf{C}_{i,\ell}).$$

This yields

$$R_{T,1} = \sum_{i \notin I^*}n_i(T)\,\Delta_i$$

$$R_{T,2} = \sum_{i=1}^{m}\sum_{\ell=1}^{n_i(T)}f_i^{\circ}(\mathbf{X}_{i,\ell}, \mathbf{C}_{i,\ell}) = \sum_{i=1}^{m}\sum_{\ell=1}^{n_i(T)}\left\{f_i^*(\mathbf{C}_{i,\ell}) - f_i(\mathbf{X}_{i,\ell}, \mathbf{C}_{i,\ell})\right\}$$

$$R_{T,3} = \sum_{i=1}^{m}\sum_{\ell=1}^{n_i(T)}\left\{\mathbb{E}[f_i^{\circ}(\mathbf{X}_{i,\ell}, \mathbf{C}_{i,\ell}) \mid \boldsymbol{f}, \mathcal{D}^{i,\ell-1}] - f_i^{\circ}(\mathbf{X}_{i,\ell}, \mathbf{C}_{i,\ell})\right\}.$$

We will upper bound the three components of the regret, $R_{T,1}$, $R_{T,2}$ and $R_{T,3}$, separately.

**Bounds on the arm selection regret $R_{T,1}$:** We have

$$R_{T,1} \leq \sum_{i \notin I^*, n_i(T) > 1} \sqrt{n_i(T)} \sqrt{n_i(T)} \Delta_i + \sum_{i=1}^m \Delta_i$$

$$\leq \sum_{i \notin I^*} \sqrt{n_i(T)} \left( \sqrt{\mathsf{C}_i \beta_i(n_i) \gamma_i(n_i)} + 2\rho_i(n_i) \right) + \sum_{i=1}^m \Delta_i$$

$$\leq \left\{ T \sum_{i \notin I^*} \mathsf{C}_i \beta_i(n_i) \gamma_i(n_i) \right\}^{1/2} + \left\{ 4T \sum_{i \notin I^*} \rho_i^2(n_i) \right\}^{1/2} + \sum_{i=1}^m \Delta_i.$$

The first inequality above is obtained by simple algebra using the fact that $\Delta_i \geq 0$ for every $i$, the second inequality follows from Lemma 4, while the third inequality is a consequence of the Cauchy-Schwarz inequality and the fact that the terms $n_i(T)$ sum to $T$. Since all the functions $\beta_i(\cdot)$ and $\rho_i(\cdot)$ are nondecreasing, we get

$$R_{T,1} \leq \sqrt{mT} \left( \sqrt{\mathsf{C}\beta(T)\bar{\gamma}(T)} + 2\rho(T) \right) + \sum_{i=1}^m \Delta_i \leq 2\bar{c}(T)\sqrt{mT} + \|\boldsymbol{\Delta}\|_1. \tag{17}$$

An alternative bound can be obtained by using Lemma 4 to infer that

$$R_{T,1} \leq \sum_{i \notin I^*} \left( \left\{ \frac{\sqrt{\mathsf{C}_i \beta_i(n_i) \gamma_i(n_i)} + 2\rho_i(n_i)}{\Delta_i} \right\}^2 + 1 \right) \Delta_i$$

$$\leq \sum_{i \notin I^*} \left\{ \frac{4c^2(n_i)}{\Delta_i} + \Delta_i \right\} \leq 4c^2(T) \sum_{i \notin I^*} \frac{1}{\Delta_i} + \|\boldsymbol{\Delta}\|_1. \tag{18}$$

**Bounds on the action selection regret $R_{T,2}$:** Let us use the shorthand notation $\lambda = \varphi\sqrt{\log(4md\psi/\delta)}$. In view of Lemma 1, on the event $\mathcal{A}(\delta) \subset \bigcap_{i=1}^m \mathcal{A}_i(\delta/2m)$, for any arm $i \in [m]$ and for any $n \in \mathbb{N}^*$, we have

$$\sum_{\ell=1}^n \left( f_i^*(\mathbf{C}_{i,\ell}) - f_i(\mathbf{X}_{i,\ell}, \mathbf{C}_{i,\ell}) \right) \leq 2 \sum_{\ell=1}^n \sqrt{\beta_i(\ell)}\, \sigma_{i,\ell-1}(\mathbf{X}_{i,\ell}, \mathbf{C}_{i,\ell}) + \frac{\lambda}{m}$$

$$\leq 2\sqrt{\beta_i(n)} \sum_{\ell=1}^n \sigma_{i,\ell-1}(\mathbf{X}_{i,\ell}, \mathbf{C}_{i,\ell}) + \frac{\lambda}{m}$$

$$\leq 2\sqrt{n\beta_i(n)} \left\{ \sum_{\ell=1}^n \sigma_{i,\ell-1}(\mathbf{X}_{i,\ell}, \mathbf{C}_{i,\ell}) \right\}^{1/2} + \frac{\lambda}{m}$$

$$\leq \sqrt{n\mathsf{C}_i \beta_i(n)\gamma_i(n)} + \frac{\lambda}{m}. \tag{19}$$

Here, for the first inequality, we used (7) and (8), for the second inequality, we used the fact that $\beta_i(\cdot)$ is a nondecreasing function, the third line follows from the Cauchy-Schwarz inequality, whereas the forth line is a consequence of inequality (10). This leads to a first bound on the second component of the regret:

$$R_{T,2} \leq \sum_{i=1}^m \sqrt{n_i \mathsf{C}_i \beta_i(n_i)\gamma_i(n_i)} + \lambda$$

$$\leq \sqrt{\mathsf{C}\beta(T)} \sum_{i=1}^m \sqrt{n_i \gamma_i(n_i)} + \rho(T)$$

$$\leq \sqrt{mT\mathsf{C}\beta(T)\bar{\gamma}(T)} + \rho(T) = \sqrt{mT}\, \bar{c}(T). \tag{20}$$

An alternative bound can be obtained by replacing in (19) $n$ by $n_i = n_i(T)$ and, for $i \notin I^*$, using Lemma 4:

$$\sum_{\ell=1}^{n_i} \left( f_i^*(\mathbf{C}_{i,\ell}) - f_i(\mathbf{X}_{i,\ell}, \mathbf{C}_{i,\ell}) \right) \leq 2\sqrt{\kappa\beta(1)} + \frac{4c^2(T)}{\Delta_i}.$$

The first term in the right hand side above comes from the fact that if $n_i = 1$, then instead of using Lemma 4, we can simply use the fact that $C_i \gamma_i(1) \leq 4\kappa_i$. Summing over all $i \in [m]$, we get

$$R_{T,2} \leq \sqrt{T|I^*|}\, c(T) + 2m\sqrt{\kappa\beta(1)} + 5c^2(T)\sum_{i\notin I^*} \frac{1}{\Delta_i}. \tag{21}$$

**Bounds on the stochastic-error-term $R_{T,3}$**   Let us denote by $\mathcal{F}_{i,n}$ the $\sigma$-algebra generated by $\{\mathcal{D}^{i,n}, \boldsymbol{f}\}$ and define $M_{i,0} = 0$ and

$$M_{i,n} = \sum_{\ell=1}^{n} \big\{ \mathbb{E}[f_i^\circ(\mathbf{X}_{i,\ell}, \mathbf{C}_{i,\ell}) \mid \boldsymbol{f}, \mathcal{D}^{i,\ell-1}] - f_i^\circ(\mathbf{X}_{i,\ell}, \mathbf{C}_{i,\ell}) \big\}, \qquad \text{for } n \in \mathbb{N}^*.$$

It is clear that $M_{i,n}$ is a $\mathcal{F}_{i,n}$-martingale, in the sense that $\mathbb{E}[M_{i,n} \mid \mathcal{F}_{i,n-1}] = M_{i,n-1}$. To upper bound $M_{i,n}$, we will apply the Azuma-Hoeffding inequality (Azuma, 1967; Hoeffding, 1963). To this end, note that $f_i^\circ(\mathbf{x}, \mathbf{c}) \geq 0$ for every $\mathbf{x}$ and $\mathbf{c}$. Furthermore, as seen in the proof of Lemma 3, the set of functions $f_i$ satisfying the inequalities

$$f_i^\circ(\mathbf{X}_{i,\ell}, \mathbf{C}_{i,\ell}) = \max_{\mathbf{x}} f_i(\mathbf{x}, \mathbf{C}_{i,\ell}) - f_i(\mathbf{X}_{i,\ell}, \mathbf{C}_{i,\ell})$$

$$\leq \Delta f_i^* \leq d'r'\varphi\sqrt{\log(4m\bar{d}\psi/\delta)} := \lambda'$$

is of probability at least $1 - \delta/2m$. This implies that conditionally to $\boldsymbol{f}$, $|M_{i,n} - M_{i,n-1}| \leq \lambda'$. Therefore, we can apply the Azuma-Hoeffding inequality to infer that

$$\mathbb{P}\big(M_{i,n} \leq \lambda'\sqrt{2n\log(1/\delta_1)} \mid \boldsymbol{f}\big) \geq 1 - \delta_1$$

with probability at least $1 - \delta/2$ over the randomness in $\boldsymbol{f}$. Choosing $\delta_1 = \delta/(2m\pi_n)$, and using the union bound, we arrive at

$$\mathbb{P}\big(M_{i,n} \leq \lambda'\sqrt{2n\log(2m\pi_n/\delta)}, \text{ for all } n \in \mathbb{N}^*,\ i \in [m] \mid \boldsymbol{f}\big) \geq 1 - \frac{\delta}{2}$$

with probability at least $1 - \delta/2$ over the randomness in $\boldsymbol{f}$. From this, we readily get

$$\mathbb{P}\big(M_{i,n} \leq \lambda'\sqrt{2n\log(2m\pi_n/\delta)}, \text{ for all } n \in \mathbb{N}^*,\ i \in [m]\big) \geq 1 - \delta.$$

Hence, on an event $\mathcal{B}(\delta)$ of probability at least $1 - \delta$, we have

$$R_{T,3} = \sum_{i=1}^{m} M_{i,n_i(T)} \leq \lambda'\sqrt{2\log(2m\pi_T/\delta)} \sum_{i=1}^{m} \sqrt{n_i(T)} \tag{22}$$

$$\leq \lambda'\sqrt{2mT\log(2m\pi_T/\delta)}. \tag{23}$$

The alternative upper bound on the term $R_{T,3}$ is obtained by combining (22) with Lemma 4. This yields

$$R_{T,3} \leq \rho(T)\Big( \sqrt{T|I^*|} + m + 2c(T)\sum_{i\notin I^*}\frac{1}{\Delta_i} \Big). \tag{24}$$

Combining upper bounds (17), (20) and (23), we get

$$R_T \leq 3\sqrt{mT}\,\bar{c}(T) + \|\boldsymbol{\Delta}\|_1 + \lambda'\sqrt{2mT\log(2m\pi_T/\delta)}$$

$$\leq 5\sqrt{mT}\,\bar{c}(T) + \|\boldsymbol{\Delta}\|_1$$

with probability at least $1 - 2\delta$.

Similarly, combining bounds (18), (21) and (24), we get

$$R_T \leq 4c^2(T)\sum_{i\notin I^*}\frac{1}{\Delta_i} + \|\boldsymbol{\Delta}\|_1 + \sqrt{T|I^*|}\,c(T) + 2m\sqrt{\kappa\beta(1)} + 5c^2(T)\sum_{i\notin I^*}\frac{1}{\Delta_i}$$

$$+ \rho(T)\Big( \sqrt{T|I^*|} + m + 2c(T)\sum_{i\notin I^*}\frac{1}{\Delta_i} \Big).$$

Regrouping the terms, we arrive at

$$R_T \leq 11c^2(T)\sum_{i\notin I^*}\frac{1}{\Delta_i} + \|\boldsymbol{\Delta}\|_1 + 2\sqrt{T|I^*|}\,c(T) + m\big(2\sqrt{\kappa\beta(1)} + \rho(T)\big).$$

This completes the proof of the proposition. $\qquad\square$

### B.3 PROOF OF THEOREM 1

We denote $\bar{\varphi} = \varphi \vee 1$ and $\bar{\psi}_= \psi \vee 1$, and take $\pi_n = \pi^2 n^2/6$. Proposition 1 assumes that

$$\beta_i(n) \geq 2\log(4/\delta) + 4d_i \log(mr_i d_i \pi_n),$$

where the right-hand side can be upper bounded by

$$2\log(4/\delta) + 4d_i \log(mr_i d_i \pi_n) \leq 4d_i \log(2\pi_n r_i md_i/\delta)$$
$$\leq 8d_i \log\left(\frac{\pi\sqrt{r_i}}{3} \cdot \frac{md_i n}{\delta}\right),$$

since $d_i \geq 1$.

Now we analyse the assumption on $\rho_i(n)$:

$$\rho_i(n) \geq \varphi\sqrt{log(4m\bar{d}_i\psi/\delta)} + d_i' r_i' \varphi \log\left(4m(d_i'\psi + 2\pi_n)/\delta\right),$$

where the right-hand side can be upper bounded by

$$\varphi\sqrt{\log(4m\bar{d}_i\psi/\delta)} + d_i' r_i' \varphi \log\left(4m(d_i'\psi + 2\pi_n)/\delta\right)$$
$$\leq \varphi\left(\log(4m\bar{d}_i\bar{\psi}/\delta) + d_i' r_i' \log\left(4m(\bar{d}_i\psi + 2\pi_n)/\delta\right)\right)$$
$$\leq 2\varphi d_i'(r_i' \vee 1)\log\left(4m\left(\frac{\bar{d}_i\bar{\psi}}{\delta} + \frac{\pi^2 n^2}{3}\right)\right)$$
$$\leq 2\varphi d_i'(r_i' \vee 1)\log\left(8m\bar{d}_i\bar{\psi}\pi^2 n^2/(3\delta)\right), \text{ as } a + b \leq 2ab, \ \forall a, b \geq 1$$
$$\leq 4\varphi d_i'(r_i' \vee 1)\log\left(\frac{2\sqrt{2}\pi\bar{\psi}}{\sqrt{3}} \cdot \frac{mn\bar{d}_i}{\delta}\right).$$

Hence, $\beta_i(n)$ and $\rho_i(n)$ satisfy the conditions of Proposition 1 if

$$\beta_i(n) \wedge \rho_i(n) \geq 8\bar{\varphi}\bar{d}_i(\bar{r}_i \vee 1)\log\left(5.14\bar{\psi}\sqrt{\bar{r}_i \vee 1} \cdot mn\bar{d}_i/\delta\right).$$

By taking $\mathsf{A}_1 = 8\bar{\varphi}\bar{\psi}(\bar{r}_i \vee 1)$, then $\beta_i(n)$ and $\rho_i(n)$ satisfy the conditions of Theorem 1 whenever they satisfy the conditions of Proposition 1.

Under Section 3 the regret is upper bounded

$$R_T \leq 5\sqrt{mT}\left(\sqrt{\mathsf{C}\beta(T)\bar{\gamma}(T)} + \rho(T)\right) + \|\mathbf{\Delta}\|_1$$
$$\leq 5\sqrt{\mathsf{C} \vee 1}\{\sqrt{mT}\left(\sqrt{\beta(T)\bar{\gamma}(T)} + \rho(T)\right)\} + \|\mathbf{\Delta}\|_1,$$

which implies $\mathsf{A}_2 = 5\sqrt{\mathsf{C} \vee 1}$.

For the instance dependent upper bound, Section 3 implies that

$$R_T \leq 2\sqrt{|I^*|T}\left(\sqrt{\mathsf{C}\beta(T)\bar{\gamma}(T)} + \rho(T)\right) + 22\left(\mathsf{C}\beta(T)\gamma(T) + \rho^2(T)\right)\sum_{i\notin I^*}\frac{1}{\Delta_i}$$

$$+ \|\mathbf{\Delta}\|_1 + 2m\left(\sqrt{\kappa\beta(1)} + \rho(T)\right)$$
$$\leq 2\sqrt{\mathsf{C} \vee 1}\{\sqrt{|I^*|T}\left(\sqrt{\beta(T)\bar{\gamma}(T)} + \rho(T)\right)\} + 22(\mathsf{C} \vee 1)\left(\beta(T)\gamma(T) + \rho^2(T)\right)\sum_{i\notin I^*}\frac{1}{\Delta_i}$$

$$+ 2(\sqrt{\kappa} \vee 1)m\left(\sqrt{\beta(1)} + \rho(T)\right) + \|\mathbf{\Delta}\|_1.$$

Consequently, we take $\mathsf{A}_3 = 2(11\mathsf{C} \vee \sqrt{\kappa} \vee 11)$.

In summary, if we take $\mathsf{A}_1 = 8\bar{\varphi}\bar{\psi}(\bar{r}_i \vee 1)$, $\mathsf{A}_2 = 5\sqrt{\mathsf{C} \vee 1}$, and $\mathsf{A}_3 = 2(11\mathsf{C} \vee \sqrt{\kappa} \vee 11)$ then Section 3 implies Theorem 1.

| Kernel | $\gamma_T$ | $\max_{n_i \in \mathbb{N}: \sum n_i = T} \sum_{i=1}^{m} \gamma_{n_i},$ |
|---|---|---|
| Matérn-$\nu$ [3] | $\mathcal{O}(T^{\frac{d}{2\nu+d}} \log^{\frac{2\nu}{2\nu+d}}(T))$ | $\mathcal{O}(m^{\frac{2\nu}{2\nu+d}} T^{\frac{d}{2\nu+d}} \log^{\frac{2\nu}{2\nu+d}}(T) + m)$ |
| Squared exponential [3] | $\mathcal{O}(\log^{d+1}(T))$ | $\mathcal{O}(m \log^{d+1}(T))$ |
| Cauchy spectral mixture [4] | $\mathcal{O}(T^{\frac{2d^2+2d+1}{2d^2+2d+2}} \log(T))$ | $\mathcal{O}(m^{\frac{1}{2d^2+2d+2}} T^{\frac{2d^2+2d+1}{2d^2+2d+2}} \log(T) + m)$ |
| $D$-dimensional feature-map [5] | $\mathcal{O}(D \log(T))$ | $\mathcal{O}(mD \log(T))$ |
| $\beta$-polynomial eigendecay [3] | $\mathcal{O}(T^{1/\beta} \log^{1-1/\beta}(T))$ | $\mathcal{O}(m^{1-1/\beta} T^{1/\beta} \log^{1-1/\beta}(T) + m)$ |
| $\beta$-exponential eigendecay [3] | $\mathcal{O}(\log^{1+1/\beta}(T))$ | $\mathcal{O}(m \log^{1+1/\beta}(T))$ |

Table 3: Bounds on maximum information gains and their sums on a domain of dimension $d$.

## C  ADDITIONAL THEORETICAL RESULTS

In this section, we provide some additional mathematical results that support the discussion conducted after Theorem 1.

### C.1  BOUNDS ON THE INFORMATION GAINS

This subsection aims to upper bound sums of the following form

$$\max_{\substack{n_i \in \mathbb{N}^+ \\ \sum n_i = T}} \sum_{i=1}^{m} \gamma_i(n_i), \tag{25}$$

where $\gamma_i$ are the maximum information gains for possibly different Gaussian processes (*i.e.*, with different kernels) but satisfying our assumptions (0 mean, covariance function is bounded by $\kappa$, etc.). We set $\gamma_i(t) = 0, \ \forall t \leq 0$ for convenience.

To upper bound (25), we use results of Vakili et al. (2021) that imply that

- for kernels with exponential eigendecay $\gamma_T \leq \mathcal{O}(\log^{1+\frac{1}{\beta_e}})$ where $\beta_e$ is a positive constant,
- for kernels with polynomial eigendecay $\gamma_T \leq \mathcal{O}(T^{\frac{1}{\beta_p}} \log^{1-\frac{1}{\beta_p}} T)$, where $\beta_p > 1$.

As an example, for widely used kernels, such Matern-$\nu$ $\gamma_T \leq \mathcal{O}(T^{\frac{d}{2\nu+d}} \log^{\frac{2\nu}{2\nu+d}} T)$ and $\gamma_T \leq \mathcal{O}(\log^{d+1} T)$ for the squared exponential kernel.

Hence, we write that $\gamma_i(T) \leq A_i T^{a_i} \log^{b_i} T + B_i$ for some $A_i, B_i, b_i > 0$ and $0 \leq a_i < 1$. For simplicity, we take $A, B, b$ and $a$ to be the maximums of the corresponding constants, e.g. $a := \max_{i \in [m]} a_i$. So we have that

$$\gamma_i(T) \leq A T^a \log^b T + B, \quad \forall i \in [m]. \tag{26}$$

We note that $B \geq \gamma_i(1), \ \forall i \in [m]$ which under the assumption that kernels are bounded by $\kappa$ is at most $\frac{1}{2} \log(1 + \sigma^{-2} \kappa)$. So $B \geq \frac{1}{2} \log(1 + \sigma^{-2} \kappa)$.

**Proposition 2.** *If the array $(\gamma_i(n))_{i \in [m], n \in \mathbb{N}^*}$ satisfies (26), then*

$$\max_{\substack{n_i \in \mathbb{N}^+ \\ \sum n_i = T}} \sum_{i=1}^{m} \gamma_i(n_i) \leq A m^{1-a} T^a \log^b T + mB.$$

---

[3] Please see Vakili et al. (2021)
[4] Please see Zhang & Hua (2025)
[5] Please see Srinivas et al. (2010)

*Proof.* Using that $x \mapsto \log^b x$ is increasing and that $x \mapsto x^a$ is concave, we upper bound (25) by

$$\max_{\substack{n_i \in \mathbb{N}^+ \\ \sum n_i = T}} \sum_{i=1}^{m} \gamma_{i,n_i} \leq A \max_{\substack{n_i \in \mathbb{R}, \, n_i \geq 1 \\ \sum n_i = T}} \log^b T \sum_{i=1}^{m} n_i^a + mB$$

$$\leq Am \log^b T \max_{\substack{n_i \in \mathbb{R}, \, n_i \geq 1 \\ \sum n_i = T}} \left( \frac{1}{m} \sum_{i=1}^{m} n_i \right)^a + mB$$

$$= Am^{1-a} T^a \log^b T + mB$$

and the claim of the proposition follows. $\qquad\square$

We illustrate the implications of Proposition 2 for several widely used kernels in Table 3. Our results also extend to composite kernels, i.e. sums and products of kernels. In particular, Theorem 3 of (Krause & Ong, 2011) gives an upper bound on the maximum information gain of a sum kernel in terms of the information gains of its components. Similarly, Theorem 2 of (Krause & Ong, 2011) shows that when the component kernels are finite-rank, the information gain of their product can be bounded likewise by those of the components.

## C.2 ON THE UNIQUENESS OF THE OPTIMAL ARM

This section motivates the uniqueness of the optimal arm in the scenario where $(f_i)_i$ are sampled from Gaussian processes. We recall that the reward of arm $i$ is

$$\mu_i = \mathbb{E}\left[ \sup_{\mathbf{x} \in \mathfrak{X}_{\mathbf{x}}} f_i(\mathbf{x}, \mathbf{C}) \mid f_i \right]. \tag{27}$$

We next show when $\mu_i \neq \mu_j$ for distinct $i$ and $j$.

**Lemma 5.** *Let $i \neq j$. If at least one of $\mu_i$ and $\mu_j$ is absolutely continuous, then $P(\mu_i = \mu_j) = 0$.*

*Proof.* Note that $\mu_i$ is $\sigma(f_i)$ measurable and $\mu_j$ is $\sigma(f_j)$ measurable. By Assumption 2, the random variables $f_i$ and $f_j$ are independent, which implies that any random variables defined on their respective $\sigma$-algebras are also independent, in particular $\mu_i$ and $\mu_j$ are independent.

We assume without loss of generality that $\mu_i$ is absolutely continuous. Under the established independence we write that

$$P(\mu_i = \mu_j) = (P_{\mu_i} \otimes P_{\mu_j})(\mu_i = \mu_j) = \int P_{\mu_i}(\mu_i = x) \, P_{\mu_j}(\mathrm{d}x)$$

$$= \int 0 \, P_{\mu_j}(\mathrm{d}x) = 0,$$

which concludes the proof. $\qquad\square$

It follows from Lemma 5 that if all $\mu_i$, except possibly one, are absolutely continuous, then almost surely no two distinct ones are equal.

The next lemma provides sufficient conditions for $\mu_i$ to be absolutely continuous.

**Lemma 6.** *Let $|\mathfrak{X}_{\mathbf{C}(i)}| < \infty$ and $k_i$ be strictly positive definite, then $\mu_i$ is absolutely continuous.*

*Proof.* Our Assumption 2 implies that $f_i$ is almost surely continuous and hence, thanks to the compactness assumption made in Assumption 1, attain their maxima on $\mathfrak{X}_{\mathbf{X}(i)}$. We may rewrite the expectation in (27) as a sum, assuming that $\mathfrak{X}_{\mathbf{C}(i)} = \{\mathbf{c}_1, \ldots, \mathbf{c}_J\}$:

$$\mu_i = \sum_{\mathbf{c} \in \mathfrak{X}_{\mathbf{C}(i)}} P(\mathbf{C} = \mathbf{c}) f_i\left(\mathbf{x}_i^*(\mathbf{c}), \mathbf{c}\right)$$

$$= \max_{\mathbf{x}_1, \ldots, \mathbf{x}_J} \sum_{j=1}^{J} \nu_j f_i(\mathbf{x}_j, \mathbf{c}_j).$$

It is clear that the process

$$F_i(\mathbf{x}_1, \ldots, \mathbf{x}_J) = \sum_{j=1}^{J} \nu_k f_i(\mathbf{x}_j, \mathbf{c}_j)$$

is centered and Gaussian. For $\vec{\mathbf{x}} = (\mathbf{x}_1, \ldots, \mathbf{x}_K)$ and $\vec{\mathbf{x}}' = (\mathbf{x}'_1, \ldots, \mathbf{x}'_K)$, we have

$$\mathbb{E}[F(\vec{\mathbf{x}})F(\vec{\mathbf{x}}')] = \boldsymbol{\nu}^{\mathsf{T}} \mathbf{K}_i(\vec{\mathbf{x}}, \vec{\mathbf{x}}')\boldsymbol{\nu}, \tag{28}$$

where $\boldsymbol{\nu}^{\mathsf{T}} = (\nu_1, \ldots, \nu_J)$ and

$$\mathbf{K}_i(\vec{\mathbf{x}}, \vec{\mathbf{x}}') = \begin{pmatrix} k_i((\mathbf{x}_1, \mathbf{c}_1), (\mathbf{x}'_1, \mathbf{c}_1)) & \ldots & k_i((\mathbf{x}_1, \mathbf{c}_1), (\mathbf{x}'_J, \mathbf{c}_J)) \\ \vdots & \ddots & \vdots \\ k_i((\mathbf{x}_J, \mathbf{c}_J), (\mathbf{x}'_1, \mathbf{c}_1)) & \ldots & k_i((\mathbf{x}_J, \mathbf{c}_J), (\mathbf{x}'_J, \mathbf{c}_J)) \end{pmatrix}.$$

Note that $\mathbf{K}_i(\vec{\mathbf{x}}, \vec{\mathbf{x}}')$ is strictly positive definite since $k_i$ is strictly positive definite. So (28) is strictly greater than zero for all $\mathbf{x}$ and $\mathbf{x}'$ from their respective domains.

We have that $F_i$ is a zero-mean Gaussian process whose index set is compact and separable, as it is a compact subset of Euclidean space, and its variance is zero nowhere. This implies that $\sup_{\mathbf{x}} F(\vec{\mathbf{x}})$ is absolutely continuous (Lifshits, 1984, Corollary, page 2), and so is $\mu_i$. $\qquad\square$

Consequently, if we have that for all $i$, except possibly one, $|\mathfrak{X}_{\mathbf{C}(i)}| < \infty$ and $k_i$ are strictly positive definite, we can invoke Lemma 6 and Lemma 5 to conclude that all $\mu_i$ are almost surely distinct. So $I^*$ set contains only one element almost surely.

