# OpenReview forum: "Contextual Causal Bayesian Optimisation"
_ICLR.cc/2026/Conference — ICLR 2026 Poster_

### Official Review · Reviewer_AJMc · 2025-10-31

**Soundness:** 3
**Presentation:** 2
**Contribution:** 3
**Rating:** 4
**Confidence:** 3

**Summary:**

The authors argue that existing Causal Bayesian Optimization and Contextual Bayesian Optimization are not suitable for problems where both causal and contextual information is present. They propose a method which combines those two methods in order to construct a method which can handle such problems.

**Strengths:**

- The paper illustrates well (albeit with a toy example) that using only context or only causal information can lead to unnecessarily large regrets.
- Despite "simply" combining Causal BO and Contextual BO, this blend is non-trivial enough for the contributions of this paper to be relevant.
- Both experimental and theoretical evidence is presented in defense of the proposed method.

**Weaknesses:**

- The introduction does not motivate the problem (where does it emerge? Any specific cases?), and the first paragraph seems disconnected from the rest, mentioning POMISs, which are not used anywhere in the text.
- Not enough is said about CaBO and CoBO for context, and for comparison with the method proposed in this paper. It should be very clear where exactly this method differs from those two.
- It is not explained why the problem/task of the agent (lines 120-127) is relevant to study in the first place, even in principle. This is my most serious problem with this paper. See questions 1-3 below about this.

**Questions:**

1. Can the authors provide an example of a situation modeled by the mathematical problem described in the box at lines 120-128?
2. Why would the practitioner/agent not use all of the context it has access to? Why should it choose any scope other than the maximal (and still informative) one at every step? Is it because mixed policies would be harder to learn or is there something else to it?
3. The past observations (line 122) do not include the elements of $\mathbf{v}_{l}$ not in $\mathbf{c}_{l}$. Is this sensible? I understand that the mixed policy only makes use of the $\mathbf{c}_{l}$, but why not make use of the observed $\mathbf{v}_{l} \setminus \mathbf{c}_{l}$ in future steps?
4. Your definition of POMP drops the requirement of them being NROs from (Lee & Barenboim 2020). What is the justification for this? This should be justified in the paper.
5. In line 133, why $\mu^{*}_{S} > \mu^{*}_{S'}$ instead of $\mu^{*}_{S} \ge \mu^{*}_{S'}$? As it stands, won't we eliminate MPS $S$ that are indeed optimal for some SCM, but it just so happens that there is another MPS $S'$ which leads to the same average reward? It seems to me that both $S$ and $S'$ should be taken as POMPS in that case, not both eliminated.
6. In line 138 you state that the regret would increase linearly. Why?
7. Isn't the MAB too "complex" to actually converge to something useful? I'll explain: the MAB has POMPS as arms, but for each POMP $\mathcal{S}$ that it may choose, there are multiple (possibly uncountable) policies $\pi_{\mathcal{S}}$ that can be chosen; hence the $\mu_{S}$ that the MAB is trying to maximize is an expected value over a complicated distribution combining the different $\pi_{\mathcal{S}}$ and the probabilities of each of them being chosen, which heavily depends on the details of the chosen BO method. Another way of viewing this, is that each arm of the MAB does not simply have one reward distribution, but multiple, and the choice of which to sample from is done by the BO. Could you explain why this is not an issue?
8. Related to the previous question: from what I gather, what we want to choose is not the best $\mathcal{S}$, but the best pair $(\mathcal{S}^{*}, \pi_{\mathbf{S}}^{*})$. It seems to me that it could easily happen that the best such pair would not be found by the proposed method. It may happen that $A$ suggests $\mathcal{S}^{*}$ a few times but a very bad $\pi_{\mathcal{S}}$ is suggested by the BO, and the MAB "concludes" that $\mu_{\mathcal{S}^{*}}$ is low and starts picking other MPSs. Thus, I do not see the algorithm being trustworthy for reasonable (i.e. non-astronomical) numbers of $T$. Could you explain why this is not an issue?
9. In the paragraph starting at line 237, the authors justified why CaBO's selection method is unsuitable for the case with context. But why is CoCa-BO's method suitable? This is not explained in the main text/in a high-level way. Could you also include a discussion of your method and why it is suitable in the main text?

### Other comments

- It should be clearer that the definition of mixed policy scope comes from (Lee & Bareinboim 2020), for example by citing in the definition's title. Same comment holds for other concepts taken from that or other sources.
- Re-defining $G_S$ in the first footnote can cause confusion. I would stick would the formally correct definition only.
- "the nature generates..." (line 123/124) is strangely phrased.
- typo line 132: ...*is* a possibly-optimal...
- In line 136, you retroactively changed the protocol in the box. This can make it confusing to understand the exact problem this paper is trying to solve.
- typo line 919: equivalence
- Defining the MAB as a mapping seems to hide the actual complexity of that step. I would suggest explaining in detail what is the MAB problem here (see also question above).
- line 209: I do not see how the details of the MAB update step are provided by Algorithm 2. Is this a typo?

---

> ### Author Response · Authors · 2025-11-21
> **Point-to-point reply to the reviewer's questions**
>
> We sincerely thank the reviewer for their careful reading and thoughtful summary of our work. We are particularly encouraged by the recognition of our efforts to provide both experimental and theoretical evidence.
>
> Below, we address the reviewer’s questions in detail. We believe that our responses and the revisions in the manuscript effectively address the weaknesses raised. Should any of our answers require further clarification, we remain fully available for discussion during the review period.
>
>
> > 1. Can the authors provide an example of a situation modeled by the mathematical problem described in the box at lines 120-128?
>
> We have revised the introduction to better motivate our setting and method. For the reviewer’s convenience, we summarize the scenarios discussed in our motivation below:
>
> 1. **Complex Environments (e.g., Video Games):**
>    Setting causality aside, consider environments like *Dota 2* [1] or *StarCraft* [2]. In these settings, the extreme complexity of the game makes it unclear how to best describe the state or actions *a priori*. Multiple plausible state and action descriptors can be defined, each corresponding to a distinct scope within our framework. While our focus is on Bayesian optimization rather than reinforcement learning, these scenarios illustrate the relevance of our approach.
>
> 2. **Portfolio Optimization:**
>    In portfolio optimization [3, 4], multiple strategies may be employed, each controlling specific sets of portfolio parameters conditioned on market observables. This setup naturally defines a multi-scope optimization problem, where the goal is to identify the optimal strategy across multiple scopes.
>
> 3. **Neural Architecture Search:**
>    As highlighted in [5], there are multiple strategies for neural architecture search. For example:
>    - One strategy might monitor gradient and activation magnitudes, adjusting network width and weight initialization.
>    - Another might monitor test-time performance and the generalization gap, adjusting learning rate, gradient clipping, and depth.
>
>    While controlling all architectural parameters and conditioning on all performance metrics might theoretically yield the best performance, the enormous search space makes this infeasible. Researchers therefore rely on heuristic strategies, which align with our framework of optimizing over multiple scopes.
>
> The causal-graph component of our framework allows for the precise and systematic construction of such strategies and their domains (i.e., POMPS), ensuring simplicity without sacrificing optimality.
>
>
> > 2. Why would the practitioner/agent not use all of the context it has access to? Why should it choose any scope other than the maximal (and still informative) one at every step? Is it because mixed policies would be harder to learn or is there something else to it?
>
> In our setting, the context $C_X$ guides the selection of an interventional value for the variable $X$. While this choice of $X$ directly influences the target $y$, it may also affect other variables in the system. However, these secondary effects, resulting from our intervention on $X$, are not essential for learning the optimal policy. Additionally, observing these variables may be costly. Therefore, we designed the setting such that observing these values is not required.
>
> In addition, a policy scope that includes all contextual variables may be redundant. Finally, it can be beneficial to consider a scope with fewer variables to reduce the sample complexity of learning the optimal policy.
>
> > 3. The past observations (line 122) do not include the elements of $\mathbf v_l \notin \mathbf c_l$.
> > Is this sensible? I understand that the mixed policy only makes use of the $\mathbf c_{l}$, but why not make use of the observed $\mathbf v_{l} \setminus \mathbf c_{l}$ in future steps?
>
> In our setting, the process unfolds in two rounds at each step:
> -  First round: We select a set of intervenable variables and a set of contextual variables to observe.
> -  Second round: After observing the contextual variables, we choose the values for the intervenable variables.
>
> There are two possible approaches for observations in the second round:
> - Option 1 (adopted in our work): Observe only the target variable.
> - Option 2: Observe all variables $\mathbf{v}_l \setminus \mathbf{c}_l$.
>
> Our proposed algorithm is designed and analyzed for the more general setting where only the target variable is observed in the second round. Additionally, observing all variables may be costly in practical applications. These considerations motivated our choice of Option 1.

---

> ### Author Response · Authors · 2025-11-21
>
> > 4. Your definition of POMP drops the requirement of them being NROs from (Lee & Barenboim 2020). What is the justification for this? This should be justified in the paper.
>
>  This was initially done to save space; however, we agree with the reviewer that it caused confusion, and we have addressed it in our revision.
>
> > 5. In line 133, why $\mu^\star_{S} > \mu^\star_{S'}$ instead of $\mu^\star_{S} \ge \mu^\star_{S'}$?  As it stands, won’t we eliminate MPSs that are indeed optimal for some SCM, but just so happen that there is another MPS which leads to the same average reward?  It seems to me that both $S$ and $S'$ should be taken as POMPS in that case, not both eliminated.
>
> Our definition is adopted from [6], which implicitly assumes a unique maximizing MPS for a given SCM. While this assumption may not hold in all cases, such as when the maximum of $\mu_{\mathcal{S}}$ is achieved by multiple MPS, we believe it is reasonable for most practical scenarios. To maintain consistency with prior work, we prefer to retain the original definition.
>
> That said, our method is compatible with any sensible modification of the POMPS definition, including cases with multiple maximizers. For instance, we could follow the reviewer’s suggestion and replace the strict inequality with $\geq$ to accommodate such scenarios.
>
>
> > 6. In line 138 you state that the regret would increase linearly. Why?
>
> This question is perhaps related to the previous discussion about the strict inequality in the definition of POMPS. The linear increase in regret arises from the possibility of discarding a potentially optimal POMPS.
>
> Consider two POMPS, $\mathcal{S}$ and $\mathcal{S'}$, derived from the causal graph. If we discard $\mathcal{S'}$, there may exist an SCM for which $\mu^\star_{\mathcal{S'}} > \mu^\star_{\mathcal{S}}$. In this case, even if the agent acts optimally according to $\mathcal{S}$, it incurs an intermediate expected regret of $\mu^\star_{\mathcal{S'}} - \mu^\star_{\mathcal{S}}$ at each step due to the exclusion of $\mathcal{S'}$.
>
> As a result, the total expected regret accumulates as $T \cdot (\mu^\star_{\mathcal{S'}} - \mu^\star_{\mathcal{S}})$, growing linearly with $T$. Of course, this is not true if the strict inequality is replaced by $\ge$.
>
> > 7. Isn't the MAB too ""complex"" to actually converge to something useful?
> > I'll explain: the MAB has POMPS as arms, but for each POMP $\mathcal{S}$ that it may choose, there are multiple (possibly uncountable) policies $\pi_{\mathcal{S}}$ that can be chosen; hence, the $\mu_{\mathcal{S}}$ that the MAB is trying to maximize is an expected value over a complicated distribution combining the different $\pi_{\mathcal{S}}$ and the probabilities of each of them being chosen, which heavily depends on the details of the chosen BO method.
> > Another way of viewing this is that each arm of the MAB does not simply have one reward distribution, but multiple, and the choice of which to sample from is done by the BO.
> > Could you explain why this is not an issue?
>
> At a high level, the effectiveness of our approach is supported by Theorem 1 and empirical results. To illustrate why this works, consider a simplified scenario with two contextual variables, $C_1$ and $C_2$, an intervenable variable $X$, and a target variable $Y$. Assume $C_1$ and $C_2$ are discrete, taking values in $\{1, \ldots, K_c\}$, and $X$ is binary. For each value of $C_1$, there are two possible intervention values, leading to $2^{K_c}$ policies. The same applies to the scope $(C_2, X)$, resulting in a total of $2 \times 2^{K_c}$ policies, each associated with a distribution of $Y$.
>
> This setup can be viewed as a MAB problem with $2 \times 2^{K_c}$ arms. One could group these arms into two sets—one for each scope—and update a single UCB per group. In the first round, the group with the highest UCB is selected. In the second round, within the chosen group, the arm with the highest individual UCB is played. Intuitively, this strategy is effective, as it reduces to selecting the arm with the highest UCB across all arms, regardless of the grouping.
>
> However, if there is additional structure within each group, we can replace the UCB strategy in the second round with a more efficient method, avoiding the exponential complexity of exhaustive search. This is exactly what Bayesian optimization is designed for. In our setting, we achieve this by assuming smoothness in the family of distributions induced by the Gaussian prior. This assumption allows us to leverage the properties of the kernel corresponding to each POMPS, as specified in Assumption 2.
>
> The exploration rate of the MAB in our algorithm, $\rho_i(n)$, depends on the dimensionality of the POMPS and the smoothness parameters of the function class induced by the Gaussian process (see line 316). This enables adaptive exploration: higher exploration rates for less-smooth function classes, and vice versa.

---

> ### Author Response · Authors · 2025-11-21
>
> > 8. Related to the previous question: from what I gather, what we want to choose is not the best $\mathcal S$, but the best pair $(\mathcal S, \pi_{\mathcal S})$.
> > It seems to me that it could easily happen that the best such pair would not be found by the proposed method.
> > It may happen that $\mathcal{S}$ is suggested by the BO, and the MAB concludes that $\mu_{\mathcal{S}}$ is low and starts picking other MPSs.
> >Thus, I do not see the algorithm being trustworthy for reasonable (i.e., non-astronomical) numbers of $T$.
> >Could you explain why this is not an issue?
>
> We believe there is a misunderstanding in the reasoning: $\mathcal{S}$ is not suggested by Bayesian optimization (BO). Instead, BO selects the interventional values of the intervenable variables, while the scope $\mathcal{S}$ is chosen by the MAB.
>
> The scenario described in the question is unlikely to occur due to the strong connection between the MAB and BO, as formalized in Equation (5). Specifically, the upper confidence bound (UCB) used in the MAB is derived from the posterior mean and variance, ensuring that the selection of $\mathcal{S}$ and the corresponding policy $\pi_{\mathcal{S}}$ are tightly coupled.
>
> Empirically, our method demonstrates validity for reasonable values of $T$. For instance, when using squared exponential kernels, a common choice in Bayesian optimization, our regret bound is $\tilde{O}(\sqrt{T})$, up to polylogarithmic terms in $T$.
>
>
> > 9. In the paragraph starting at line 237, the authors justified why CaBO's selection method is unsuitable for the case with context. But why is CoCa-BO's method suitable? This is not explained in the main text/in a high-level way. Could you also include a discussion of your method and why it is suitable in the main text?
>
>
> The CoCa-BO selection criterion is suitable because it tracks historical context values and their corresponding upper confidence intervals (line 936 of Algorithm 2). The averaging in line 936 upper bounds the reward from a POMPS when played according to that POMPS’s optimal policy.
>
> Additionally, $\bar U_{i_t} (n_{i_t})$ shrinks quickly enough that the algorithm does not select suboptimal POMPS too often, as shown by Lemma 4.
>
> Following reviewer's recommendation, we added a short discussion in the main text (end of Section 3).
>
>
> ### **Other comments**
> > 1. It should be clearer that the definition of mixed policy scope comes from (Lee & Bareinboim 2020), for example by citing in the definition's title. Same comment holds for other concepts taken from that or other sources.
>
> You are right. On line 110, we added "The following definitions and terminology are based on Lee and Bareinboim (2020)."
>
> >2. Re-defining in the first footnote can cause confusion. I would stick would the formally correct definition only.
>
> Thank you for your remark. We removed the footnote.
>
> >3.  "the nature generates..." (line 123/124) is strangely phrased.
>
> We agree. We changed it to "the environment generates $\mathbf v_t$ according to the
> distribution $P_{\mathbf{V}}^{\pi_t}$".
>
> > 4. typo line 132: ...is a possibly-optimal...
>
> Thank you, we corrected the typo.
>
> > 5. In line 136, you retroactively changed the protocol in the box. This can make it confusing to understand the exact problem this paper is trying to solve.
>
> Thank you for raising this point. We did not change the protocol; rather, we clarified the scope of our analysis. Specifically, we restrict our attention to policies that select \(\mathcal{S}\) from \(\mathbb{S}[\mathcal{G}]\), as any choice outside this set would yield suboptimal results. We have rephrased the paragraph to make this clearer and hope this resolves any confusion.
>
>
> > 6. typo line 919: equivalence
>
> Thank you, we corrected the typo.
>
> > 7. Defining the MAB as a mapping seems to hide the actual complexity of that step. I would suggest explaining in detail what is the MAB problem here (see also question above).
>
> Please refer to our response to the question.
>
> > 8. line 209: I do not see how the details of the MAB update step are provided by Algorithm 2. Is this a typo?
>
> Line 936 of the submitted version was describing the MAB update, but we agree with you that the definition of UCB was not given in Algorithm 2. It was presented in Eq (5). In the revised version, we explicitly refer to Eq. (5) in Algorithm 2.
>
> ### **Cited references**
>  **[1]** Dota 2 with Large Scale Deep Reinforcement Learning, OpenAI et al, 2019.
>
>  **[2]** Grandmaster level in StarCraft II using multi-agent reinforcement learning, Vinyals, O., Babuschkin, I., Czarnecki, W.M. et al., Nature 2019.
>
>  **[3]** Portfolio optimization in stochastic markets, U. Çakmak and S. Özekici, Springer 2005
>
>  **[4]** Bayesian Optimization of Risk Measures, Sait Cakmak et. al., NeurIPS 2020
>
>  **[5]** Neural Architecture Search: A Survey, Thomas Elsken et. al., JMLR 2019
>
>  **[6]** Characterizing Optimal Mixed Policies: Where to Intervene and What to Observe, Sanghack Lee, Elias Bareinboim, NeurIPS 2020.

---

### Official Review · Reviewer_xqZh · 2025-10-31

**Soundness:** 3
**Presentation:** 3
**Contribution:** 3
**Rating:** 6
**Confidence:** 4

**Summary:**

The paper proposes CoCa-BO, a framework that unifies causal and contextual Bayesian optimisation by integrating the concept of Mixed Policy Scopes (MPSs) from causal inference with Bayesian optimisation (BO) methods. The algorithm operates in two layers: (i) a multi-armed bandit (MAB) mechanism that adaptively selects among possibly-optimal mixed policy scopes (POMPSs), and (ii) a Gaussian-process-based BO routine that optimises interventions within each chosen scope. Theoretical analysis establishes high-probability sublinear regret bounds for this hierarchical process. Empirical studies on synthetic causal models demonstrate that CoCa-BO avoids linear regret suffered by standard CaBO and CoBO baselines in regimes where either causality or context alone is insufficient.
Strengths

**Strengths:**

Summary
The paper proposes CoCa-BO, a framework that unifies causal and contextual Bayesian optimisation by integrating the concept of Mixed Policy Scopes (MPSs) from causal inference with Bayesian optimisation (BO) methods. The algorithm operates in two layers: (i) a multi-armed bandit (MAB) mechanism that adaptively selects among possibly-optimal mixed policy scopes (POMPSs), and (ii) a Gaussian-process-based BO routine that optimises interventions within each chosen scope. Theoretical analysis establishes high-probability sublinear regret bounds for this hierarchical process. Empirical studies on synthetic causal models demonstrate that CoCa-BO avoids linear regret suffered by standard CaBO and CoBO baselines in regimes where either causality or context alone is insufficient.

**Weaknesses:**

Weaknesses
Limited empirical validation

Experiments are mostly synthetic or semi-realistic; no real-world dataset or ablation studies.

The runtime overhead of managing multiple scopes is not quantified.

Independence across scopes

Each POMPS maintains a separate GP, which may underutilise shared structure among overlapping interventions.

Discussion on potential information sharing (e.g., multitask GPs or shared hyperpriors) would be valuable.

Algorithmic detail gaps

Algorithm 1 is schematic; explicit update formulas for Opts.update() and A.update() are missing.

Schedules for \beta_t,\rho_t and constants in the bounds are not provided, hindering reproducibility.

Clarity
Generally clear and well-written, though dense. A figure illustrating the two-level UCB mechanism (scope-level vs. within-scope) would enhance intuition. Captions could be more self-contained. Also the authors could reintroduce POMPS and MPS for better readability.
Relation to Prior Work
Missing citations include:

Aglietti et al. (2020) Multi-task Causal Learning with Gaussian Processes (very relevant in my opinion)
Lattimore et al. (2016) Causal Bandits: Learning good interventions via Causal Inference

Chowdhury & Gopalan (2017)  On Kernelised multi armed bandits

Adding these would help situate the work fully.

**Questions:**

see above

---

> ### Author Response · Authors · 2025-11-21
> **Point-to-point reply to the reviewer's concerns (1/2)**
>
> We sincerely thank the reviewer for their careful reading and thoughtful summary of our work.
>
> Below, we address the weaknesses highlighted by the reviewer. Should any of our answers require further clarification, we remain fully available to engage in discussion during the review period.
>
> > Experiments are mostly synthetic or semi-realistic; no real-world dataset or ablation studies.
>
> Given the paper’s significant theoretical contributions, our experimental evaluation aligns with the standards commonly reported in the literature on Bayesian optimization with causal or contextual information. Unfortunately, there are few widely accepted benchmarks for evaluating methods like ours.
>
> We would greatly appreciate any suggestions from the reviewer regarding datasets or additional baselines that could further demonstrate the generalizability of our approach. If specific datasets or baselines are recommended, we are happy to incorporate them into our evaluation.
>
> To address some of the other reviewers' concerns, we have also conducted additional experiments, which are reported in our responses and in the revised manuscript.
>
>
> > The runtime overhead of managing multiple scopes is not quantified.
>
> The overall runtime of our online learning algorithm up to horizon $T$ is $O(T^3)$. For example, if there are two scopes, one selected $a$ times and the other $(T-a)$ times by time $T$, the runtime, dominated by Gaussian process posterior updates for both scopes, is $a^3 + (T-a)^3 \leq T^3$. This reasoning extends to $K > 2$ scopes: the total runtime, $\sum_{i=1}^K a_i^3$ (where $\sum_{i=1}^K a_i = T$), remains bounded by $T^3$. Thus, the runtime order is the same as using a single policy scope over the entire horizon $T$.
>
>
> > Each POMPS maintains a separate GP, which may underutilise shared structure among overlapping interventions;
> Discussion on potential information sharing (e.g., multitask GPs or shared hyperpriors) would be valuable.
>
> We appreciate your suggestion regarding the potential to exploit shared information across POMPS. Following your recommendation, we have added a discussion on this point at the end of Section 5.
>
> Our current design, where each scope maintains its own contextual GP trained only on samples observed when that scope was active, avoids imposing restrictive assumptions about structure or similarity between scopes.
>
> For example, consider two policy scopes, $\mathcal{S}$ and $\mathcal{S'}$, each with one unique interventional variable but otherwise identical contextual and interventional variables. The optimal policies and target functions, $f_{\mathcal{S}}$ and $f_{\mathcal{S}'}$, may differ entirely, meaning knowledge from one scope does not necessarily transfer to the other. Introducing structured learning, such as defining a kernel over POMPS to reflect relationships between $f_{\mathcal{S}}$ and $f_{\mathcal{S}'}$, would require a kernel on varying domains and sets of random variables. However, such a kernel could be irregular and violate the smoothness assumptions typical in Bayesian optimization.
>
> Our method makes no assumptions about the distribution of contextual variables or a priori structure between POMPS. We agree that identifying assumptions to enable structured learning, without compromising the generality of our approach, is an important direction for future work.
>
>
>
> > Algorithm 1 is schematic; explicit update formulas for Opts.update() and A.update() are missing.
> > Schedules for \beta_t,\rho_t and constants in the bounds are not provided, hindering reproducibility.
>
> Algorithm 1 is indeed intended to provide a schematic for our proposed method while the details are encapsulated in Algorithm 2. The explicit values of $A_1, A_2, A_3$ are derived in Appendix B.3, and consequently the values for $\beta_t$ and $\rho_t$ are defined per Theorem 1, in line 316.

---

> ### Author Response · Authors · 2025-11-21
> **Point-to-point reply to the reviewer's concerns (2/2)**
>
> > Generally clear and well-written, though dense. A figure illustrating the two-level UCB mechanism (scope-level vs. within-scope) would enhance intuition. Captions could be more self-contained. Also the authors could reintroduce POMPS and MPS for better readability.
>
> We appreciate the reviewers' feedback and the suggestion of illustrating the two-level UCB mechanism to enhance the intuition of the method. We included this figure on page 4 of the revised manuscript. We also note that Definition 1 (line 101) and Definition 3 (line 132) reintroduce POMPS and MPS to improve the paper’s coherence.
>
>
> > Relation to Prior Work Missing citations include:
> >
> > - Aglietti et al. (2020) Multi-task Causal Learning with Gaussian Processes (very relevant in my opinion)
> > - Lattimore et al. (2016) Causal Bandits: Learning good interventions via Causal Inference.
> > - Chowdhury & Gopalan (2017) On Kernelised multi armed bandits
>
>
>
> We thank the reviewer for the references. We include a discussion of [1,2] in Section 5 (Related Work) of the revised version. Below is a brief summary:
>
> [1] analyses when information sharing across tasks is possible, modelling tasks as non-contextual interventions on distinct sets of variables. Their results show that information sharing is possible if and only if no variable confounded with the target simultaneously has unconfounded incoming and outgoing edges. Unlike [1], we make no structural assumptions about the causal graph and consider contextual interventions, to which their results do not apply. We also note that their information-sharing mechanism requires evaluating complex integrals, which is computationally demanding. Designing computationally efficient and provably correct information-sharing mechanisms without stringent structural assumptions remains an open problem, even for non-contextual interventions.
>
> [2] estimates the effect of actions on the target variable using a mixture of passive observations and guided interventions, and they establish favourable regret bounds for their approach. Unlike us, they assume no unobserved confounding and that all variables have finite support. Moreover, their interventions are non-contextual.
>
> [3] does not address causal optimisation; instead, it proposes improvements to GP-UCB [Srinivas 2009] by better controlling the exploration rate in the frequentist setting. It does not target hierarchical online learning, as in our work, where we learn both the optimal POMPS and the policy, nor the challenges introduced by such nested optimisation. Moreover, our setup is Bayesian in nature.
>
>
> ### **References**
>
> **[1]** Multi-task Causal Learning with Gaussian Processes, Aglietti et al. (2020)
>
> **[2]** Causal Bandits: Learning good interventions via Causal Inference, Lattimore et al. (2016)
>
> **[3]** On Kernelised multi armed bandits, Chowdhury & Gopalan (2017)

---

### Official Review · Reviewer_Y6Wq · 2025-11-02

**Soundness:** 2
**Presentation:** 3
**Contribution:** 2
**Rating:** 4
**Confidence:** 3

**Summary:**

The authors introduce a unified framework for contextual and causal Bayesian optimization (CoCa-BO) that aims to design intervention policies maximizing the expected value of a target variable. The proposed approach integrates both observed contextual information and known causal graph structures to effectively guide the search process. Theoretical contributions include worst-case and instance-dependent high-probability regret bounds for the algorithm. Empirical evaluations across diverse environments further demonstrate that the proposed approach achieves sublinear regret and significantly reduces sample complexity in high-dimensional settings.

**Strengths:**

The paper presents an extensive theoretical analysis of the proposed method. In addition, the authors use real-world examples to illustrate and support the proposed method.

**Weaknesses:**

Although the paper provides thorough theoretical results on the regret bounds, several important aspects require further clarification and experimental validation:

(1) Missing Baselines.
 The experimental evaluation lacks comparisons with several relevant baselines, such as existing causal Bayesian optimization (CaBO) methods — e.g., CBO (Aglietti et al., 2020) and MCBO (Sussex et al., 2023). Even though the proposed framework targets a specific problem setting, existing CBO methods could potentially be applied and should therefore be included for a fair comparison. Moreover, standard Bayesian optimization (BO) baselines such as UCB-based methods should be evaluated as well, since in some scenarios UCB can outperform CBO approaches.

(2) Limited Discussion of Related Work.
The paper should provide a more comprehensive discussion of prior work on Contextual Bayesian Optimization (CoBO) and include corresponding experimental comparisons. While the authors compare CoCa-BO against CoBO, they do not discuss other existing CoBO methods in depth, leaving the relationship between these approaches unclear.

(3) Unclear Distinction from Existing CBO Frameworks.

The conceptual and methodological differences between existing CaBO and the proposed CoCa-BO remain insufficiently discussed. If existing CaBO methods are capable of incorporating contextual variables, the motivation for introducing CoBO and CoCa-BO becomes less convincing. For instance, in the illustrative example shown in Figure 1, the context variable C could also be integrated within most CaBO frameworks.
To strengthen the paper’s contribution, the authors should include additional experiments comparing CoCa-BO with CaBO models that explicitly incorporate contextual variables.

**Questions:**

(a) The advantages  of CoCa-BO over CaBO with contextual modeling remains unclear. The authors should provide additional experimental evidence to explicitly demonstrate the advantages of CoCa-BO over CaBO when context variables are considered.

(b) The experimental evaluation would benefit from including more baselines and diverse datasets to better substantiate the generalizability of the proposed method.

(c) Minor issue: the spelling should be standardized to “Bayesian optimization” (instead of optimisation).

---

> ### Author Response · Authors · 2025-11-21
> **Point-to-point reply to the reviewer's concerns and questions (1/2)**
>
> We sincerely thank the reviewer for their careful reading and thoughtful summary of our work. We are particularly encouraged by the recognition of our efforts to provide theoretical results and illustrations on the examples from real world.
>
> Below, we address the weaknesses highlighted by the reviewer and provide detailed responses to the questions raised in a separate section. Should any of our answers require further clarification, we remain fully available to engage in discussion during the review period.
>
> #### **Weaknesses**
> > (1) Missing Baselines. The experimental evaluation lacks comparisons with several relevant baselines, such as existing causal Bayesian optimization (CaBO) methods — e.g., CBO (Aglietti et al., 2020) and MCBO (Sussex et al., 2023). Even though the proposed framework targets a specific problem setting, existing CBO methods could potentially be applied and should therefore be included for a fair comparison. Moreover, standard Bayesian optimization (BO) baselines such as UCB-based methods should be evaluated as well, since in some scenarios UCB can outperform CBO approaches.
>
>
> Our experimental evaluation does include a causal Bayesian optimization baseline, referred to as **CaBO** in our paper [1]. We did not evaluate **MCBO** due to its highly restrictive assumptions, which are not satisfied in our experimental settings. Additionally, MCBO’s constraints on interventions make it incompatible with more general methods like ours or CaBO, limiting the fairness of a direct comparison. We discuss this further in Section 5.
>
> We also included a UCB-based Bayesian optimization baseline: contextual BO with a UCB acquisition function [2], referred to as CoBO in our experiments. Notably, our method itself is UCB-based (see Algorithm 2) and leverages CoBO as a subroutine.
>
> > (2) Limited Discussion of Related Work. The paper should provide a more comprehensive discussion of prior work on Contextual Bayesian Optimization (CoBO) and include corresponding experimental comparisons. While the authors compare CoCa-BO against CoBO, they do not discuss other existing CoBO methods in depth, leaving the relationship between these approaches unclear.
>
> In the originally submitted version, we focused the discussion on the most relevant prior work due to space constraints. We have now extended this discussion by adding a paragraph in Section 5.
>
> Our evaluation centers on contextual Bayesian optimization (CoBO) with a UCB-based next evaluation point selection method [2]. The UCB approach is widely used in practice and theoretically proven to achieve optimality in many scenarios [3], which motivated its use as a baseline. Our proposed method also leverages UCB, with key modifications tailored to our framework (see Algorithm 2).
>
> While we recognize the potential value of exploring alternative CoBO implementations or conducting ablation studies, our paper already includes a comprehensive set of experiments and theoretical results that validate the effectiveness of our approach. Given the paper’s density, we prioritized clarity and depth in presenting our method’s performance.
>
> > (3) Unclear Distinction from Existing CBO Frameworks.
> The conceptual and methodological differences between existing CaBO and the proposed CoCa-BO remain insufficiently discussed. If existing CaBO methods are capable of incorporating contextual variables, the motivation for introducing CoBO and CoCa-BO becomes less convincing. For instance, in the illustrative example shown in Figure 1, the context variable C could also be integrated within most CaBO frameworks. To strengthen the paper’s contribution, the authors should include additional experiments comparing CoCa-BO with CaBO models that explicitly incorporate contextual variables.
>
> Our contribution, CoCa-BO, is distinct from CoBO (contextual BO). CoBO optimizes over fixed policy scopes, which, as demonstrated in Section 2 and our experiments (Figure 2(d)), leads to suboptimal performance. This aligns with theoretical results from [4], which show that fixed policy-scope optimization is inherently suboptimal.
>
> Existing CaBO frameworks either disregard contextual information entirely or impose restrictive assumptions on the causal graph and allowable interventions. We provide a detailed discussion of these limitations in Section 5 of our revision. The most closely related work, CaBO [1], treats all non-interventional variables as unobserved and marginalizes them out, effectively ignoring contextual information.
>
> Simply replacing the standard GP in CaBO with a contextual GP is not sufficient, as it leads to unstable scope switching during optimization. This occurs because CaBO compares scopes using acquisition values evaluated at different context values, which hinders convergence. To address this, we introduced a new criterion for selecting among POMPS (see line 936).

---

> ### Author Response · Authors · 2025-11-21
> **Point-to-point reply to the reviewer's concerns and questions (2/2)**
>
> #### **Questions**
> >(a) The advantages of CoCa-BO over CaBO with contextual modeling remains unclear. The authors should provide additional experimental evidence to explicitly demonstrate the advantages of CoCa-BO over CaBO when context variables are considered.
>
> In Section 5, Figure 2(b) demonstrates that our method, which leverages contextual information, outperforms CaBO, which does not. Additionally, Appendix A.1.2 shows that context is essential for achieving optimality. A similar example is provided in Figure 1 and its accompanying discussion.
>
> If the reviewer has specific experiments in mind that could further highlight the advantages of CoCa-BO over CaBO, we would be happy to conduct these experiments and share the results during the discussion period.
>
>
> > (b) The experimental evaluation would benefit from including more baselines and diverse datasets to better substantiate the generalizability of the proposed method.
>
> Given the paper’s significant theoretical contributions, our experimental evaluation aligns with the standards commonly reported in the literature on Bayesian optimization with causal or contextual information. Unfortunately, there are few widely accepted benchmarks for evaluating methods like ours.
>
> We would greatly appreciate any suggestions from the reviewer regarding datasets or additional baselines that could further demonstrate the generalizability of our approach. If specific datasets or baselines are recommended, we are happy to incorporate them into our evaluation.
>
> To address some of the reviewers' concerns, we have also conducted additional experiments, which are reported in our responses and in the revised manuscript.
>
>
> #### **Cited references**
>
>  **[1]** Causal Bayesian Optimization, Aglietti et. al., AISTATS 2020
>
>  **[2]** Contextual Gaussian Process Bandit Optimization, Andreas Krause, Cheng S. Ong, NeurIPS 2011
>
>  **[3]** On the Sublinear Regret of GP-UCB, Justin Whitehouse, Zhiwei Steven Wu, Aaditya Ramdas, NeurIPS 2023
>
>  **[4]** Characterizing Optimal Mixed Policies: Where to Intervene and What to Observe, Sanghack Lee, Elias Bareinboim, NeurIPS 2020

---

### Official Review · Reviewer_fPRW · 2025-11-03

**Soundness:** 3
**Presentation:** 3
**Contribution:** 2
**Rating:** 4
**Confidence:** 4

**Summary:**

The paper introduces CoCa-BO, a unified framework that optimises intervention policies while jointly selecting the policy scope (which variables to intervene on and which contexts to condition on) using causal structure and observed context. It generalises and connects Causal BO (CaBO), which uses a known DAG but ignores context, Contextual BO (CoBO) which uses context but fixes scope showing that either alone can suffer linear regret in plausible settings. CoCa-BO treats each possibly-optimal mixed policy scope (POMPS) as an arm in a bandit over scopes, and within a chosen scope runs contextual GP-BO (HEBO) to select the intervention values. The authors provide worst-case and instance-dependent high-probability regret bounds and experiments demonstrating sublinear regret and lower sample complexity in high-dimensional problems.

**Strengths:**

- The paper presents a clean framework that truly unifies causal scope selection with contextual action choice and identifies precise failure modes of CaBO/CoBO.
- The authors also provide sound theory with practical knobs. That is, there are clear regret guarantees with interpretable dependence on info gain; the approach also uses well-known GP-UCB machinery.
- Scope-as-arms with HEBO inside each arm is sensible and parallelisable;
- The paper is careful about the acquisition mismatch in contextual settings.
- There are also compelling empirical examples which show when each baseline fails and how CoCa-BO succeeds;  The work also includes a large dimensional environment in A.3 to stress sample efficiency.

**Weaknesses:**

- The approach requires known causal structure; while standard in CaBO, many real tasks need robustness to graph misspecification or partial knowledge. (They defer discovery to future work.)
- Computing the POMPS set can be exponential in |V| (albeit parallelisable). For very large graphs, this may become the bottleneck.
-The MAB over scopes uses historical acquisition values; bandit feedback is influenced by the within-scope BO’s learning speed and noise, which may cause slow scope identification when arms differ mainly via context distributions.
- While reasonable, results depend on a single contextual BO backend; it would strengthen claims to show backend-agnostic performance (e.g., GPflow/BoTorch comparisons)
- Robustness is explored but limited; at very high noise or with heavy-tailed outcomes, the GP assumptions might need adaptation.
- Most importantly, the approach  treats context as an observable input variable but does not explicitly model how context affects the causal mechanisms or intervention effects (e.g., through structural equations). This limits causal interpretability since the GP simply learns correlations between c_t and outcomes rather than a causal modulation of intervention efficiency.
- Each POMPS scope maintains its own contextual GP, trained on samples observed when that scope was active. This means data from different scopes, which might share the same contexts, is not shared across arms. In practice, this can lead to data inefficiency: if two scopes operate under overlapping context distributions, each must relearn context-outcome relationships independently.

**Questions:**

1. In large-dimensional settings, how does the model avoid the curse of dimensionality when embedding context directly into the GP input? Were any dimensionality-reduction or feature-selection methods considered?

2. For graphs with many nodes, enumerating POMPS can be exponential. Could you discuss heuristic or approximate methods for pruning scopes without losing theoretical guarantees?

3. Can you show a task where the same interventions yield different optima under distinct contexts, demonstrating that context truly modulates causal effects?

4. Have you compared against meta-learning or multi-task BO baselines that can share context information across tasks, to isolate the benefit of causal reasoning?

5. What happens if you remove context from the GP input but keep the same scope structure? Does performance drop substantially, confirming that context contributes non-trivially?

---

> ### Author Response · Authors · 2025-11-21
> **Point-to-point reply to the reviewer's concerns and questions (1/3)**
>
> We sincerely thank the reviewer for their careful reading and thoughtful summary of our work. We are particularly encouraged by the recognition of our efforts to unify causal scope selection with contextual action, as well as the acknowledgment of our sound mathematical results.
>
> Below, we address the weaknesses highlighted by the reviewer and provide detailed responses to the questions raised in a separate section. Should any of our answers require further clarification, we remain fully available to engage in discussion during the review period.
>
> ### **Weaknesses**
> > Computing the POMPS set can be exponential in |V| (albeit parallelisable). For very large graphs, this may become the bottleneck. -The MAB over scopes uses historical acquisition values; bandit feedback is influenced by the within-scope BO’s learning speed and noise, which may cause slow scope identification when arms differ mainly via context distributions.`
>
> The computational cost of POMPS set depends on the number of unobserved confounders. For instance, it is efficiently computable for fully connected graphs without unobserved confounders by recursively traversing the target’s parents and assigning them to the interventional or contextual set until dependence is achieved. Performance also improves significantly when few variables are confounded with the target (Prop 5, [1]). We acknowledge, as noted in our paper, that POMPS’s worst-case complexity remains exponential in $|\mathbf{V}|$, with no known algorithm offering a better worst-case guarantee.
>
> > While reasonable, results depend on a single contextual BO backend; it would strengthen claims to show backend-agnostic performance (e.g., GPflow/BoTorch comparisons).
>
> Both HEBO and BoTorch rely on GPyTorch for GP modeling, while GPflow uses its own TensorFlow-based implementation. We selected HEBO for its superior performance, as it has been shown to outperform even more modern GP bandit algorithms like Vizier [2]. In Appendix A.5, we benchmark BoTorch and GPflow, demonstrating that our method is backend-agnostic, though HEBO still performs slightly better.
>
> We include a benchmark of BoTorch and GPFlow into our revision in Appendix A.5 and show that our method is backend-agnostic, though HEBO performs slightly better. The table below provides a brief overview of the results:
>
>
> | iteration | method   | regret |
> |:---------:|----------|-------:|
> | **200** | **HEBO** | **0.110 ± 0.023** |
> | 200 | BoTorch | 0.147 ± 0.025|
> | 200 | GPflow | 0.165 ± 0.026|
> | **400** | **HEBO** | **0.069 ± 0.013** |
> | 400 | BoTorch | 0.092 ± 0.014|
> | 400 | GPflow | 0.01 ± 0.015|
> | **600** | **HEBO** | **0.056 ± 0.009** |
> | 600 | BoTorch | 0.072 ± 0.009|
> | 600 | GPflow | 0.077 ± 0.01|
> | **700** | **HEBO** | **0.051 ± 0.007** |
> | 700 | BoTorch | 0.065 ± 0.008|
> | 700 | GPflow | 0.07 ± 0.009|
> > Robustness is explored but limited; at very high noise or with heavy-tailed outcomes, the GP assumptions might need adaptation.
>
> We agree that adapting the GP assumption to handle high noise or heavy-tailed outliers is a promising direction for future research.

---

> ### Author Response · Authors · 2025-11-21
> **Point-to-point reply to the reviewer's concerns and questions (2/3)**
>
> > Most importantly, the approach treats context as an observable input variable but does not explicitly model how context affects the causal mechanisms or intervention effects. This limits causal interpretability since the GP simply learns correlations between c_t and outcomes rather than a causal modulation of intervention efficiency.
>
> Indeed, our approach does not require learning the entire causal mechanism governing variable interactions. Instead, it learns a mapping from $\mathbf C_{\mathcal{S}}$ to $\mathbf X_{\mathcal{S}}$ that optimizes the target expectation, akin to classical Bayesian optimization, which focuses on identifying promising regions of the domain rather than estimating the objective function everywhere. This can be seen as a limitation in some applications, but it is also a strength in others.
>
> The process we use provides causal explainability: the learned dependence is not merely a correlation with $c_t$, but a result of interventions that directly influence outcomes under the observed context. For a given scope $\mathcal{S}$, our agent modifies the causal mechanism generating $\mathbf X_{\mathcal{S}}$ at each time $t$. The GP learns a policy from these interventions, ensuring that when $\mathbf X_{\mathcal{S}}$ is controlled as prescribed, $\mathbb{E}[Y_t]$ is optimized.
>
> For example, consider a thermostat regulating room temperature based on outdoor conditions. The agent does not need to learn the full mechanism by which outdoor temperature and heating affect the room temperature. Instead, it only requires an optimal mapping between these inputs and the heating action. While modeling the entire mechanism is possible, it typically requires more interactions with the environment, as it demands a globally accurate approximation.
>
>
> > Each POMPS scope maintains its own contextual GP, trained on samples observed when that scope was active. This means data from different scopes, which might share the same contexts, is not shared across arms. In practice, this can lead to data inefficiency: if two scopes operate under overlapping context distributions, each must relearn context-outcome relationships independently.
>
> We acknowledge the reviewer’s point about the potential for data inefficiency due to the lack of knowledge sharing across POMPS scopes. While each scope maintains its own contextual GP trained only on samples observed when that scope was active, this design avoids imposing restrictive assumptions about structure or similarity between scopes.
>
> For example, consider two policy scopes, $\mathcal{S}$ and $\mathcal{S'}$, where each has one unique interventional variable but otherwise identical contextual and interventional variables. The optimal policies and target functions $f_{\mathcal{S}}$ and $f_{\mathcal{S}'}$ may still differ entirely, meaning knowledge from one scope does not necessarily transfer to the other. Introducing structured learning (such as defining a kernel over POMPS to reflect relationships between $f_{\mathcal{S}}$ and $f_{\mathcal{S}'}$) would require a kernel on varying domains and sets of random variables. However, such a kernel could be irregular and violate the smoothness assumptions typical in Bayesian optimization.
>
> Our method makes no assumptions about the distribution of contextual variables or a priori structure between POMPS. We agree that identifying assumptions to enable structured learning (without compromising the generality of our approach) is an important direction for future work.

---

> > ### Author Response · Authors · 2025-11-21
> > **Point-to-point reply to the reviewer's concerns and questions (3/3)**
> >
> > ### **Questions**
> >
> > > 1. In large-dimensional settings, how does the model avoid the curse of dimensionality when embedding context directly into the GP input? Were any dimensionality-reduction or feature-selection methods considered?
> >
> > Knowledge of the causal graph allows us to reduce the dimensionality of the problem by considering only the relevant variables. An example is the environment in Appendix A.3.
> > One may further reduce the effective dimension by using low-rank kernels. However, as reviewer "xqZh" noted, the paper is already dense, and space limitations prevented us from expanding in that direction.
> >
> >
> > > 2. For graphs with many nodes, enumerating POMPS can be exponential. Could you discuss heuristic or approximate methods for pruning scopes without losing theoretical guarantees?
> >
> > Prop 5 of [1] provides a criterion for pruning the causal graph without sacrificing optimality. This accelerates the computation of POMPS but does not reduce their number. It is not possible, a priori, to prune POMPS without risking optimality, by definition.
> >
> > On the positive side, our theoretical results [Lemma 4] show that our method chooses POMPS that are suboptimal for the given problem (i.e., the specific SCM compatible with the causal graph) increasingly rarely. For example, if all kernels are squared exponential, then a suboptimal POMPS is played $O($poly $\log T)$ times, where $T$ is the horizon.
> >
> >
> > > 3. Can you show a task where the same interventions yield different optima under distinct contexts, demonstrating that context truly modulates causal effects?
> >
> > The SCM in Figure 1 of our paper provides exactly such a case: the optimal policy depends non-trivially on the contextual variable. The discussion following Figure 1 explains why.
> >
> > Additionally, Figure 2 (b,c,d) reports results for policies that utilize contextual variables. We show in Appendix A.1.2 and Appendix A.2 that one must use the contextual variables to perform optimally.
> >
> > > 4. Have you compared meta-learning or multi-task BO baselines that can share context information across tasks, to isolate the benefit of causal reasoning?
> >
> > Causal information, given via causal graphical model, allows one to reduce the number of scopes being considered. If causal information is ignored, one has to consider all the possible policy scopes which are exponential in terms of variables. This results in an extremely high number of tasks, even when the number of variables is small or the causal graph is simple, which limits the application of meta-learning or multi-task BO.
> >
> > > 5. What happens if you remove context from the GP input but keep the same scope structure? Does performance drop substantially, confirming that context contributes non-trivially?
> >
> > We first note that if there are no contextual variables, our method reduces to Causal BO [3]. Specifically, the set of POMPS reduces to the set of POMIS [4], which are precisely the scopes over which CBO optimizes. Furthermore, our scope selection criterion in line 936 reduces to the causal acquisition function used in CBO. Thus, our method is an extension of CBO.
> >
> > In Figure 2b, CBO performs worse than CoCa-BO because the latter utilizes contextual information. If we drop contextual information, we would perform exactly as CBO. This confirms the crucial role of context.
> >
> > We further reinforce this by adding the requested experiment in our revision in Appendix A.5, where we remove context from the GP input while keeping the scope structure unchanged. The table below provides a brief overview of the results:
> >
> >
> > | iteration | method   | regret |
> > |:---------:|----------|-------:|
> > | **200** | **with context** | **0.110 ± 0.023** |
> > | _200_ | _w/o context_ | _0.215 ± 0.019_|
> > | **400** | **with context** | **0.069 ± 0.013** |
> > | _400_ | _w/o context_ | _0.188 ± 0.012_ |
> > | **600** | **with context** | **0.056 ± 0.009** |
> > | _600_ | _w/o context_ | _0.179 ± 0.01_ |
> > | **700** | **with context** | **0.051 ± 0.007** |
> > | _700_ | _w/o context_ | _0.176 ± 0.09_ |
> >
> > ### **Cited references**
> >
> >   **[1]**   Characterizing Optimal Mixed Policies: Where to Intervene and What to Observe, Sanghack Lee, Elias Bareinboim, NeurIPS 2020.
> >
> >   **[2]**    The Vizier Gaussian Process Bandit Algorithm, Xingyou Song, Qiuyi Zhang, Chansoo Lee, Emily Fertig, Tzu-Kuo Huang, Lior Belenki, Greg Kochanski, Setareh Ariafar, Srinivas Vasudevan, Sagi Perel, Daniel Golovin. arXiv:2408.11527 (2024)
> >
> >   **[3]**    Causal Bayesian Optimization, Aglietti et. al., AISTATS 2020
> >
> >   **[4]**    Structural Causal Bandits: Where to Intervene?, Sanghack Lee, Elias Bareinboim, NeurIPS 2018

---

### Author Response · Authors · 2025-12-02
**Summary of the rebuttal and the revised manuscript**

We sincerely thank all reviewers for their constructive and insightful feedback. We are encouraged that all reviewers acknowledged the significance of our theoretical contributions, the strength of our empirical results, and the clarity of our exposition. Despite this broadly positive assessment, three reviewers rated the paper *marginally below acceptance*, while one rated it *marginally above*. We respectfully believe that several of the key strengths highlighted in the textual reviews were not fully reflected in the final scores, and that a number of the concerns raised stemmed from misunderstandings that we have now carefully addressed.

To assist the area chair, we provide below a concise summary of the major revisions made to the paper and the main questions we resolved during the rebuttal. All substantial changes appear in **blue** in the revised manuscript for ease of verification. We hope that these clarifications help convey that the concerns raised have been fully and satisfactorily addressed.

---

## Modifications in the Revised Manuscript

- **Clarified motivation**: The introduction has been revised to more clearly articulate the relevance and importance of our setting and methodology.
- **Added intuitive schematic**: A new **Figure 2** (Section 3) offers a schematic summary of our algorithm, helping readers build intuition for our approach.
- **Justified POMPS selection**: We added a focused discussion at the end of Section 3 explaining why our POMPS selection strategy is appropriate and how it differs from existing methods.
- **Strengthened related work**: Section 5 now includes an extended discussion on information sharing between POMPS and the conditions under which it becomes feasible.
- **New ablation studies**:
  - An ablation in Appendix A.5 demonstrates that our method is backend-agnostic.
  - Another ablation highlights the critical role of contextual information.
- **Improved clarity and accuracy**: We corrected minor issues, streamlined sentences, and improved coherence throughout.

---

## Major Questions Addressed During the Rebuttal

- **Computational cost of POMPS**: We clarified that the complexity depends on the structure and number of unobserved confounders, and highlighted speedups enabled by Proposition 5 of [1], which we use in our implementation.
- **Independence from GP implementation**: We provided evidence that our method's performance does not rely on a specific GP implementation and explained our choice of HEBO as a strong state-of-the-art Bayesian optimization framework.
- **Necessity of learning the full causal model**: We explained why full SCM identification is not required for policy optimization, drawing parallels with classical BO and Bayesian experimental design, and included a motivating example.
- **Limits of information sharing across POMPS**: We emphasized that information sharing requires additional structural assumptions, consistent with observations from [2] in context-free settings.
- **Distinction from prior approaches**: We further clarified how our framework differs from previous work, supported by our baseline choices.
- **Runtime considerations**: We showed that managing multiple GPs introduces no additional runtime overhead; the only cost stems from computing POMPS.
- **Redundancy of certain scopes**: As demonstrated in Appendix A.3, restricting attention to scopes containing all contextual variables is unnecessary.
- **Two-level optimization insight**: Beyond our theoretical results, we provided an illustrative example showing why our method performs well despite the inherent nested optimization.

---

## References

[1] *Characterizing Optimal Mixed Policies: Where to Intervene and What to Observe*, S. Lee and E. Bareinboim, NeurIPS 2020.
[2] *Multi-task Causal Learning with Gaussian Processes*, V. Aglietti et al., 2020.

---

### Meta-Review · Area_Chair_rUvg · 2026-01-06

**Summary:**

Reviewer fPRW had some concerns and questions, the most notable one as far as I can see is that the proposed method treats each context separately which may lead to data inefficiencies.

Reviewer Y6Wq mainly seems to be concerned with missing baselines.

Reviewer xqZh also raised the concern that algorithm does not utilize structural connections between different interventions. But they have decided to support acceptance.

Reviewer AJMc provided an excellent review both critiquing the lack of motivation and justification for the proposed method, but also giving very technical feedback to the authors.

**Reviewer Concerns:**

I beleive the authors addressed almost all concerns of Reviewer fPRW.

Reviewer Y6Wq's concerns have been correctly addressed by the authors.

Some of Reviewer AJMc's concerns would be outstanding after the rebuttal, and I believe they would have liked to see another round of review with changes implemented by the authors.

**Reviewer Scores:**

Reviewer fPRW might have been convinced.

I am not sure if Reviewer Y6Wq would have been satisfied and raised their score but I am satisfied with the authors' rebuttal and their decision to not include some baselines due to their restrictive assumptions.

The authors did a good job addressing Reviewer AJMc's concerns, but I think it would not be enough to change their current score into the accept territory.

---

### Decision · Program_Chairs · 2026-01-26

Accept (Poster)